# Dynamic recurrence risk and adjuvant chemotherapy benefit prediction by ctDNA in resected NSCLC

Bin Qiu[1,2,4], Wei Guo[1,2,4], Fan Zhang[1], Fang Lv[1], Ying Ji[1], Yue Peng[1], Xiaoxi Chen[3], Hua Bao[3], Yang Xu[3], Yang Shao[3], Fengwei Tan[1,2], Qi Xue[1,2], Shugeng Gao[1,2✉] & Jie He[1]

Accurately evaluating minimal residual disease (MRD) could facilitate early intervention and personalized adjuvant therapies. Here, using ultradeep targeted next-generation sequencing (NGS), we evaluate the clinical utility of circulating tumor DNA (ctDNA) for dynamic recurrence risk and adjuvant chemotherapy (ACT) benefit prediction in resected non-small cell lung cancer (NSCLC). Both postsurgical and post-ACT ctDNA positivity are significantly associated with worse recurrence-free survival. In stage II-III patients, the postsurgical ctDNA positive group benefit from ACT, while ctDNA negative patients have a low risk of relapse regardless of whether or not ACT is administered. During disease surveillance, ctDNA positivity precedes radiological recurrence by a median of 88 days. Using joint modeling of longitudinal ctDNA analysis and time-to-recurrence, we accurately predict patients' post-surgical 12-month and 15-month recurrence status. Our findings reveal longitudinal ctDNA analysis as a promising tool to detect MRD in NSCLC, and we show pioneering work of using postsurgical ctDNA status to guide ACT and applying joint modeling to dynamically predict recurrence risk, although the results need to be further confirmed in future studies.

[1] Department of Thoracic Surgery, National Cancer Center/National Clinical Research Center for Cancer/Cancer Hospital, Chinese Academy of Medical Sciences and Peking Union Medical College, Beijing, China. [2] Key Laboratory of Minimally Invasive Therapy Research for Lung Cancer, Chinese Academy of Medical Sciences, Beijing, China. [3] Geneseeq Research Institute, Nanjing Geneseeq Technology Inc, Nanjing, China. [4] These authors contributed equally: Bin Qiu, Wei Guo. ✉email: gaoshugeng@vip.sina.com

Lung cancer is one of the most common malignancies and the leading cause of cancer-related deaths worldwide[1]. Non-small cell lung cancer (NSCLC) accounts for ~85% of lung cancer cases[2], and surgical resection is a preferred curative treatment for ~30% of NSCLC with stage I–IIIA disease[3,4]. After completion of definitive treatment, routine post-treatment surveillance with regular clinical assessments and radiological imaging is recommended to detect recurrent disease or metastasis. However, patients may harbor minimal residual disease (MRD)[5,6], a potential source for subsequent early recurrence and metastasis, which cannot be reliably detected by traditional radiological imaging due to its limited resolution.

Adjuvant chemotherapy (ACT) after surgery could potentially eliminate MRD and improve survival[7] and is recommended for patients with resected stage II and III NSCLC[3,4]; however, recent meta-analyses suggested only minimal benefit of adjuvant chemotherapy on 5-year survival, overall survival, and disease-free survival after complete resection in patients with early-stage NSCLC[7,8]. Nevertheless, drug toxicity has been one of the major concerns of adjuvant chemotherapy, and NSCLC patients usually suffered from several grades III and IV side effects after adjuvant chemotherapy, such as neutropenia, anemia, asthenia, nausea, vomiting, and treatment-related deaths[9]. Therefore, to reduce futile toxicities due to overtreatment, identifying reliable biomarkers to select appropriate patients for adjuvant chemotherapy is of paramount importance.

Circulating tumor DNA (ctDNA) is a promising biomarker for non-invasive molecular profiling and its ability to identify MRD and monitor recurrence has been shown in breast cancer[10,11], colorectal cancer[12], gastric cancer[13], and urothelial bladder carcinoma[14]. In lung cancer, recent studies utilizing personalized mutation detection panels or CAPP-Seq have demonstrated the potential of ctDNA for MRD detection and early detection of recurrence for patients after surgery with or without chemotherapy[15,16] and for patients with long-term responses to PD-L1 blockade[17]. Nevertheless, given that these lung cancer studies mainly focused on landmark time points and binary ctDNA detection status, the utility of using serial ctDNA changes

during disease surveillance for dynamically predicting patients' recurrence risk has not been well characterized. Furthermore, previous studies only focused on the MRD detection after the completion of adjuvant therapy to evaluate its efficacy, whereas whether ctDNA-defined MRD status could guide the administration of ACT in lung cancer patients is still unclear.

In recent years, joint modeling of longitudinal and time-to-event data has exhibited its promising utility in clinical trial studies[18,19]. Specifically, the joint modeling brings longitudinal data (e.g., circulating tumor cells or ctDNA status) and time-to-event data (e.g., recurrence-free survival or overall survival) simultaneously into a single model to improve inference for dependence and association between the longitudinal biomarker and time-to-event[20,21], which could be potentially used for dynamic postoperative recurrence risk prediction to further improve disease surveillance.

In this study, we sought to evaluate the utility of ultradeep targeted next-generation sequencing (NGS) on serial longitudinal ctDNA sampling post-surgery in resected NSCLC patients for MRD detection and ACT treatment decision-making, as well as dynamic risk prediction of recurrence during post-surgical and post-ACT disease surveillance.

## Results

**Patient characteristics and sample genomic profiling.** A total of 116 NSCLC patients who received surgical resection were enrolled in the study, 13 of whom discontinued due to various reasons and were thus excluded from subsequent analyses (Fig. 1). The clinicopathological characteristics of the remaining 103 patients were summarized in Table 1 with details presented in Supplementary Data 1. Seventy-one of 103 (68.9%) patients received ACT, including 69 patients with chemotherapy and 2 patients with chemoradiotherapy, and the median duration of ACT was 73 days (range: 5–153 days) (Supplementary Data 1). Seven *EGFR*-positive patients received either adjuvant targeted therapy or chemotherapy plus targeted therapy, while 25 patients did not receive any adjuvant treatment after surgery (Supplementary Data 1). The resected tumor tissue samples and serial

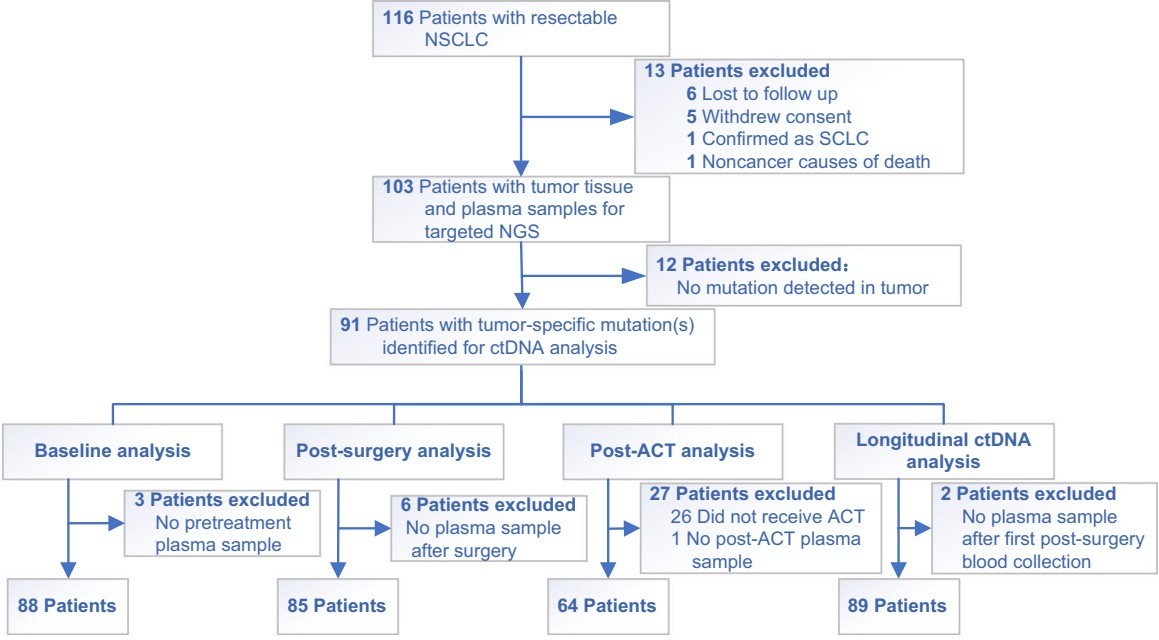

**Fig. 1 Patient enrollment flowchart.** Of the 116 lung cancer patients enrolled in our study, 103 of them had their tumor and plasma samples sequenced. After excluding patients who had no detectable tumor mutations, tumor and plasma data from 91 patients were subjected to further analyses, including 88 patients with pretreatment plasma samples, 85 with postsurgical plasma samples, 64 with post-ACT plasma samples, and 89 with serial plasma samples.

### Table 1 Patient clinical characteristics.

| Characteristics | All patients ($N = 103$) |
|---|---|
| *Age (years)* | |
| Median (range) | 64 (38–82) |
| *Gender (%)* | |
| Male | 67 (65%) |
| Female | 36 (35%) |
| *Smoking status (%)* | |
| Yes | 61 (59%) |
| No | 42 (41%) |
| *Differentiation (%)* | |
| Info available | 19 (18%) |
| Not available | 84 (82%) |
| *Histology (%)* | |
| Adenocarcinoma | 60 (58%) |
| Squamous cell carcinoma | 38 (37%) |
| Atypical carcinoid | 1 (1%) |
| Adenosquamous carcinoma | 1 (1%) |
| Large cell neuroendocrine carcinoma | 3 (3%) |
| *pTMN stage (%)* | |
| I | 12 (10%) |
| IIb | 41 (40%) |
| IIIa | 48 (47%) |
| IV | 2 (2%) |
| *T stage (%)* | |
| T1–T3 | 95 (92%) |
| T4 | 8 (8%) |
| *N stage (%)* | |
| N0–N1 | 68 (66%) |
| N2 | 35 (34%) |
| *Adjuvant therapy (%)* | |
| Chemotherapy | 72 (70%) |
| Chemoradiotherapy | 1 (1%) |
| Targeted therapy | 5 (5%) |
| Chemotherapy + targeted therapy | 2 (2%) |
| No | 23 (22%) |
| *Recurrence (%)* | |
| No recurrence | 66 (64%) |
| Locoregional recurrence | 11 (10%) |
| Lymph mode | 8 (8%) |
| lung | 7 (7%) |
| Brain | 3 (3%) |
| Other | 8 (8%) |

plasma samples at various time points during disease surveillance were collected and analyzed, with the detailed sample collection procedure described in the "Methods" section and Supplementary Fig. 1.

Genomic DNA from tumor tissues and ctDNA from plasma samples were prepared and analyzed by targeted NGS using a pre-designed lung cancer tracking panel (139 critical lung cancer-related genes) to a mean coverage depth of ~850× for tumor tissues and ~30,000× deep sequencing for ctDNA samples, followed by somatic variants analysis using Automated Triple Groom Sequencing (ATG-Seq) technology (see "Methods"). Genomic DNA from the white blood cells of the buffy coat after plasma separation was also analyzed as the normal control sample for germline variants and clonal hematopoiesis mutation filtering. ctDNA positivity was defined by accessing the presence of one or more plasma mutations that were also identified in the matched tumor sample.

**Mutational profile of tumor tissues and pretreatment ctDNA shedding**. Somatic mutations were detected in 91 of the 103 patients' tumor specimens, with a median of 2 mutations per patient (range: 1–8 mutations) (Supplementary Fig. 2). *TP53* (67/

91, 73.6%) was the most frequently mutated gene in all these NSCLC patients, which was mutated in 100% (32/32) of squamous cell carcinoma (SqCC) patients compared to 60.7% (34/56) in adenocarcinoma (AD). *EGFR*, *KRAS* mutations, and *ALK* fusions were detected in 55.4%, 16.1%, and 8.9% of AD, respectively, but not in SqCC patients, whereas *CDKN2A* (34.4% vs 5.4%), *KEAP1* (31.3% vs 3.6%), *RB1* (15.6% vs 0%), and *NFE2L2* (12.5% vs 0%) alterations were more enriched in SqCC patients. These mutational profiles are consistent with the previously reported somatic mutation landscape in East Asian NSCLC patients[22].

The pretreatment plasma samples of 88 out of the 91 patients with tumor somatic mutations detected (three patients did not have pretreatment plasma samples) were further analyzed to identify potential clinicopathological determinants of ctDNA shedding in NSCLC patients. 69.3% (61/88) of patients had detectable somatic mutations in their pretreatment plasma samples, with a median maximum variant allele frequency (VAF) of 0.34% (Supplementary Data 2). As expected, pretreatment ctDNA shedding was associated with the pTMN stage, and ctDNA detection rate in stage I/II and stage III cases was 61.0% (25/41) and 76.1% (35/46), respectively (Supplementary Fig. 3a). ctDNA shedding was also associated with histological subtype. Except for AD patients whose ctDNA-positive rate was only 49.1% (26/53 patients), 100% of patients ($n = 35$) with other NSCLC subtypes, including SqCC patients ($n = 32$), had detectable somatic mutations in the pretreatment plasma samples (Supplementary Fig. 3b). Although different histology (SqCC vs AD) showed no difference in relapse-free survival ($p = 0.52$), when only considering pretreatment ctDNA positive patients, SqCC patients shown significantly better RFS than those of AD patients (HR: 0.45; 95% CI: 0.2–0.99; $p < 0.05$).

**ctDNA positivity at landmark time points for prognosis**. To assess whether ctDNA positivity after surgery correlated with disease recurrence, ctDNA analysis was performed on post-surgical plasma samples (collected within 1-month post-surgery and before the start of ACT). Among the 85 patients who had available postsurgical plasma samples, 18 of them (21.2%; 18/85) still had detectable ctDNA with a median maximum VAF of 0.02%. Although different histology subtypes have distinct ctDNA detection rates in presurgical plasma samples (Supplementary Fig. 3b), there was no difference ($p = 1.0$) in the postsurgical ctDNA positive rate between AD (19.2%; 10 out of all the 52 AD patients) and SqCC (19.4%, 6 out of all the 31 SqCC patients). Intriguingly, these ctDNA positive patients had a significantly reduced RFS compared with ctDNA negative patients (HR: 4.0; 95% CI: 2.0–8.0; $p < 0.001$; Fig. 2a). Other risk factors, such as baseline *TP53* status (mutant vs wild type; HR: 3.3; 95% CI: 1.2–9.4; $p < 0.05$) and T stage (T4 vs T1-3; HR: 2.7; 95% CI: 1.1–6.5; $p < 0.05$), were also significantly associated with RFS (Supplementary Fig. 4 and Supplementary Table 1). We further performed survival analysis in patients with AD (Supplementary Fig. 5a) or SqCC (Supplementary Fig. 5b) separately, and their postsurgical ctDNA-positive status was still associated with worse RFS for both AD and SqCC with similar HR (8.33 and 8.56, respectively). When including postsurgical ctDNA status, histology, baseline *TP53* status, and T stage in a multivariate Cox regression analysis, postsurgical ctDNA status still had the strongest independent association with RFS ($p < 0.001$; Supplementary Table 1). Remarkably, postsurgical ctDNA positivity remained to have a significant association with worse RFS in patients with or without ACT (Supplementary Fig. 6), implying that the prognostic value of postsurgical ctDNA was independent of the administration of adjuvant therapy.

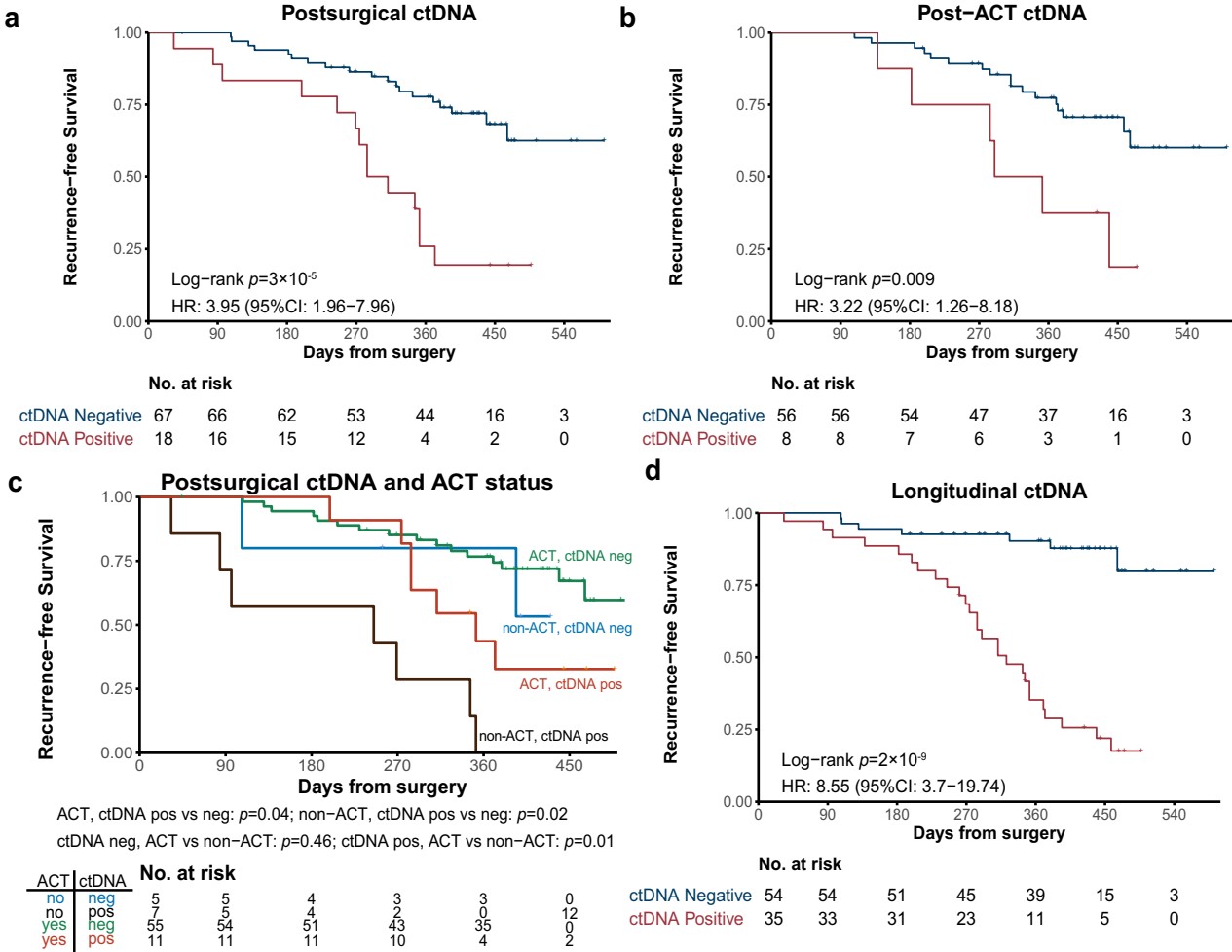

**Fig. 2 ctDNA positivity could potentially serve as a prognostic marker and guide ACT treatment. a** Kaplan–Meier curve of recurrence-free survival (RFS) in patients stratified by postsurgical ctDNA status. *p*-value was calculated by the log-rank test. **b** Kaplan–Meier curve of RFS in patients stratified by post-ACT ctDNA status. *p*-value was calculated by the log-rank test. **c** Kaplan–Meier curve of RFS in stage II–III patients stratified by both ACT treatment and postsurgical ctDNA status. *p*-value was calculated by the log-rank test for each comparison without adjustments. **d** Kaplan–Meier curve of RFS in patients stratified by longitudinal ctDNA status. *p*-value was calculated by the log-rank test.

We further explored whether ctDNA positivity after completion of ACT was associated with treatment outcome. Of the patients treated with ACT, 64 patients had their post-ACT plasma samples collected within 4 months of treatment completion and 8 (12.5%) of them were ctDNA positive with a median maximum VAF of 0.11%. Consistent with the results of postsurgical ctDNA, post-ACT ctDNA positivity was also significantly associated with worse RFS as tested by Kaplan–Meier analysis (HR, 3.2; 95% CI, 1.3–8.2; *p* < 0.05; Fig. 2b) and multivariate Cox regression analysis (*p* < 0.05; Supplementary Table 2), suggesting that ctDNA status may have potential clinical values in evaluating the effectiveness of ACT. A similar trend was observed when analyzing AD and SqCC patients separately (Supplementary Fig. 5c, d), although the results did not reach statistical significance likely due to the small sample size after stratifying patients based on the tumor types. Taken together, these results demonstrate that MRD detected by plasma ctDNA after definitive therapy is a promising prognostic biomarker for resectable NSCLC patients.

**Potential of using postsurgical ctDNA status to guide ACT decision-making.** Currently, decision-making for ACT treatment is mainly based on risk stratification of stage and other clinical factors. In the current clinical settings, stage II-III resectable NSCLC patients are considered high-risk population and are recommended for adjuvant therapy[3,4]. However, a significant proportion of these patients who received ACT still developed recurrence. We hypothesized that those patients who were clinically defined as high-risk populations but had no detectable postsurgical ctDNA may not benefit from additional ACT treatment. We therefore aimed to investigate whether postsurgical ctDNA status could identify patients who were truly benefit from ACT treatment and help guide ACT treatment in resectable lung cancer in order to improve patients' outcomes and life quality by avoiding the toxicity of ineffective ACT treatment.

In this study, the majority of stage II–III patients (84.6%; 66/78) in our cohort had ACT after surgery, except for 12 patients who did not receive ACT due to their personal unwillingness (*n* = 11) or past medical history (1 patient previously diagnosed with cerebral infarction). Within 12 non-ACT patients, all 7 patients with positive postsurgical ctDNA relapsed within one year, and 3 out of 5 patients with no detectable MRD remained recurrence-free (Supplementary Fig. 7). We stratified all the stage II-III clinical high-risk patients based on whether they received ACT treatment or whether they had detectable postsurgical ctDNA (Fig. 2c). Consistent with our hypothesis, ctDNA-positive patients had a significantly higher risk of recurrence compared to

ctDNA-negative patients within either ACT group ($p < 0.05$) or non-ACT group ($p < 0.05$). ctDNA-negative patients have a similar low risk of relapsing, independent of whether or not ACT was administered ($p = 0.46$). In contrast, ctDNA-positive patients who were treated with ACT had a significantly improved RFS than ctDNA-positive patients without ACT ($p < 0.05$). Similar results were observed when analyzed in AD and SqCC datasets separately (Supplementary Fig. 8). Therefore, postsurgical ctDNA status could potentially stratify these clinically high-risk NSCLC patients into two groups, the ctDNA-positive patients who could more likely benefit from ACT treatment, and the ctDNA-negative group where ACT seems to be unnecessary with a minimal improvement in reducing their relapse risk.

**Longitudinal ctDNA analysis for disease monitoring and relapse prediction.** We next investigated whether using longitudinal ctDNA analysis approach during the postsurgical disease surveillance can serve as a dynamic biomarker for more accurate recurrence monitoring. 79% (27/34) of relapsed patients detected at least one positive ctDNA during disease surveillance, compared with only 41% (14/34) with positive postsurgical ctDNA. This indicated that longitudinal ctDNA monitors identified more relapsed patients and are necessary for disease surveillance. Our results showed that patients with detectable ctDNA at any time point(s) during posttreatment surveillance had significantly lower DFS than those who always had negative ctDNA detection after surgery (HR, 8.5; 95% CI, 3.7–20; $p < 0.001$; Fig. 2d). In contrast, postsurgical ctDNA positivity was associated with worse RFS than negativity with a lower hazard ratio (HR: 4.0; 95% CI: 2.0–8.0; $p < 0.001$; Fig. 2a). Similar results were obtained in both AD and SqCC subgroups with similar HR (8.33 and 8.56, respectively; Supplementary Fig. 5e, f). Consistent with these results, longitudinal ctDNA status was maintained a significant association with RFS in multivariable Cox proportional hazards regression analysis along with other clinicopathological variables (HR, 7.4; 95% CI, 3.1–18; $p < 0.001$; Supplementary Table 3).

Of patients with radiological recurrence, ctDNA analysis was positive at or before relapse in 82.1% (23/28) of patients with detectable pretreatment ctDNA shedding and in 60% (3/5) of patients without pretreatment ctDNA shedding (Fig. 3a). The median lead time from ctDNA-positive detection to radiological recurrence was 88 days (Wilcoxon signed-rank test; $p < 0.001$; Fig. 3b). As an example, patient P017's CT scan 4 was considered negative for recurrence, who eventually developed bone metastasis as shown in CT scan 5 (Fig. 3c). However, ctDNA analysis has already shown as positive at CT scan 4, which was 189 days prior to the clinical imaging confirmation. Patient P072 was initially considered to be radiological recurrence at scan 4–5, but the diagnosis was corrected as no disease relapse at scan 6 (Fig. 3d, upper panel). Interestingly, the ctDNA analysis of the patient P072 stayed negative since surgery (Fig. 3d, lower panel). These results suggest that longitudinal ctDNA analysis could potentially help clarify equivocal radiological diagnosis and be a useful complement for routine clinical imaging.

**Serial ctDNA monitoring for personalized dynamic recurrence risk prediction.** When evaluating the longitudinal changes of VAFs of detected ctDNA mutations in serial plasma samples during disease surveillance, we observed a correlative trend between the ctDNA VAF changes and times to radiological recurrence in patients with ctDNA positivity at multiple time points (Supplementary Fig. 9), suggesting that the changes of ctDNA level in plasma may correlate to the risk of MRD to macroscopic (radiological) recurrence. Therefore, we sought to explore whether dynamic changes in ctDNA level can be used to predict the risk of recurrence in real-time during postsurgical surveillance of resected NSCLC.

The joint modeling is a sophisticated framework to analyze datasets when repeated measurements and time-to-event outcomes are strongly correlated[20]. It comprises two linked sub-models, one for the longitudinal process (a linear mixed sub-model of repeated measurements of ctDNA) and one for the time-to-relapse data (a Cox sub-model with baseline covariates) (Fig. 4a). It exploits the full information of collected data during follow-up up to landmark time point and takes into account the sporadic measurement error of the longitudinal data[23]. In contrast, the traditional landmark Cox model for dynamic risk prediction[24–26] applies a survival model based solely on the last observed value of the biomarker at the landmark time and obtains survival probabilities from a Cox model fitting to the patients who are still at risk at the time point of interest. Previous studies showed that hazard rates peaked at ~9 months after surgery in NSCLC patients, so we chose the closest time spot in our study (i.e., 8 months post-surgery) as one of the landmark time points, which corresponds to the time of completion of the third post-surgery blood collection. In order to evaluate which model has the best prognostic performance, we constructed both the joint model and the landmark Cox model using the information up to the landmark time and predicted the probabilities of recurrence occurring at 12 months and 15 months post-surgery. A static Cox model, which allows the prediction of event probabilities at the same time point from a fixed baseline (postsurgical ctDNA status), was also constructed for comparison. To assess the model predictive performance, we examined the discrimination power and calibration using time-dependent areas under the receiver-operating characteristics curves (AUROC) and prediction error (PE), respectively, as proposed by Rizopoulos et al[26]. A repeated fivefold cross-validation was conducted to avoid overestimation of the predictive performance.

Our results indicated that the joint model has a superior ability to predict recurrence status at 12 months (Wilcoxon rank test, $p < 0.001$; AUROC = 0.89) and 15 months (Wilcoxon rank test, $p < 0.001$; AUROC = 0.83) post-surgery, comparing with the static Cox model using only postsurgical ctDNA status, or the landmarking ctDNA status Cox model (Fig. 4b; Supplementary Fig. 10; Supplementary Table 4). Furthermore, the prediction errors of the joint model were significantly lower than those of both the landmark Cox model and static Cox model at 12 months (Fig. 4b). For 15 months, the joint model had a similar prediction error with the landmark Cox model, but significantly lower prediction errors than the static Cox model. To further estimate the calibration of the model, we constructed the joint model and landmark Cox model using the leave-one-out cross-validation method and evaluated the calibration by reliability diagrams using Hosmer-Lemeshow (H-L) test[27]. As shown in Supplementary Fig. 11, the joint model outperformed the landmarking Cox model with larger $P$ values and smaller H-L C-statistic values at both 12- and 15-month periods. In addition, joint modeling of the survival data of ADC patients alone also behaved better than the Cox method (Supplementary Fig. 12), although its predictive ability remained undetermined in SqCC patients due to the small sample size.

Dynamic recurrence risk prediction for representative patients was shown in Fig. 4c, d with personalized prediction results of other patients with two or more blood collection time points post-surgery were shown in Supplementary Fig. 13. For patient P057 (Fig. 4c), the recurrence-free probability curve did not show significant changes due to her ctDNA levels were low and stable across the three time points. In contrast, patient P017 had a faster rate of growth in the ctDNA level and had a dramatic decline in

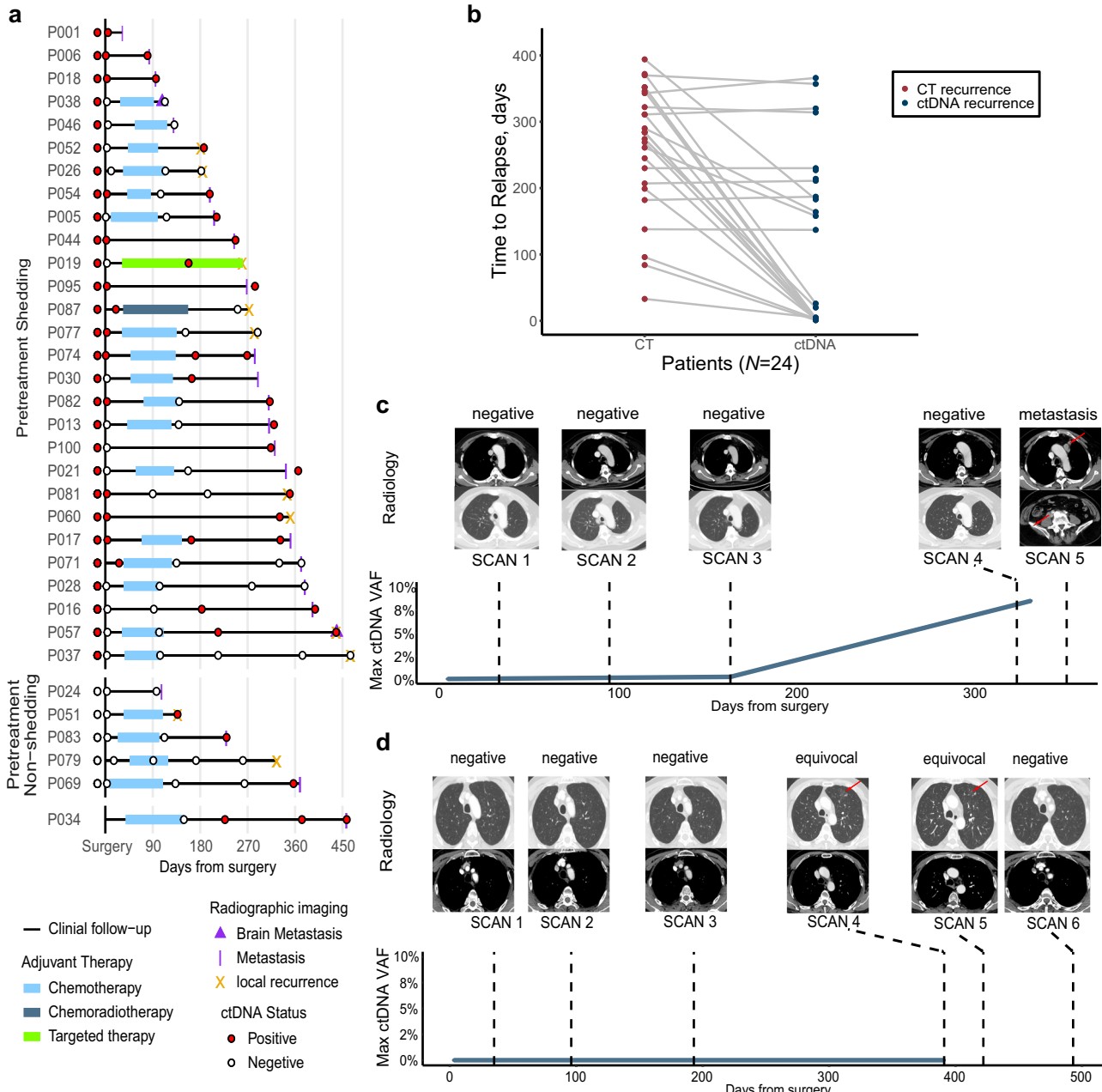

**Fig. 3 Longitudinal ctDNA analysis for relapse anticipation and disease monitoring. a** The dynamic monitoring of ctDNA in patients with radiological recurrence. Circles represented ctDNA status. Treatment and imaging information was indicated for each patient. Patients were separated based on their pretreatment ctDNA shedding status. **b** The comparison of the recurrence time measured by ctDNA versus computed tomography (CT). Two-sided Wilcoxon two-sample paired signed-rank test, $p < 0.001$. **c** CT scan and ctDNA detection for patient P017. **d** CT scan and ctDNA detection for patient P072.

the recurrence-free probability, especially after the third time point (Fig. 4d), suggesting that she was continually at increased risk of recurrence and indeed relapsed shortly after the last plasma follow-up. Taken together, these results indicated that the joint model outperforms the traditional Cox methods in disease recurrence risk prediction and could accurately estimate personalized recurrence risks for resectable NSCLC patients postsurgery using longitudinal ctDNA surveillance.

## Discussion
ctDNA monitoring has emerged as a promising approach for early diagnosis, prognosis prediction, and postsurgical surveillance. Multiple studies have demonstrated that ctDNA positivity

after surgical resection was correlated with poor patients outcomes in lung cancer[15,16,28]. In this prospective cohort study, we examined 103 NSCLC patients treated with surgical resection with or without adjuvant therapy to evaluate the utility of ctDNA in disease monitoring and treatment determination.

The detection rate of pretreatment ctDNA in our study was similar to that of previous studies[16,29], and a persistent correlation between ctDNA status of the serial plasma samples and the clinical disease course was observed, suggesting the reliability of our ctDNA detection technique. As for patients who had no recurrence despite having detectable ctDNA, three of them had positive ctDNA before or during ACT and the other two had positive ctDNA at last several time points which indicated that

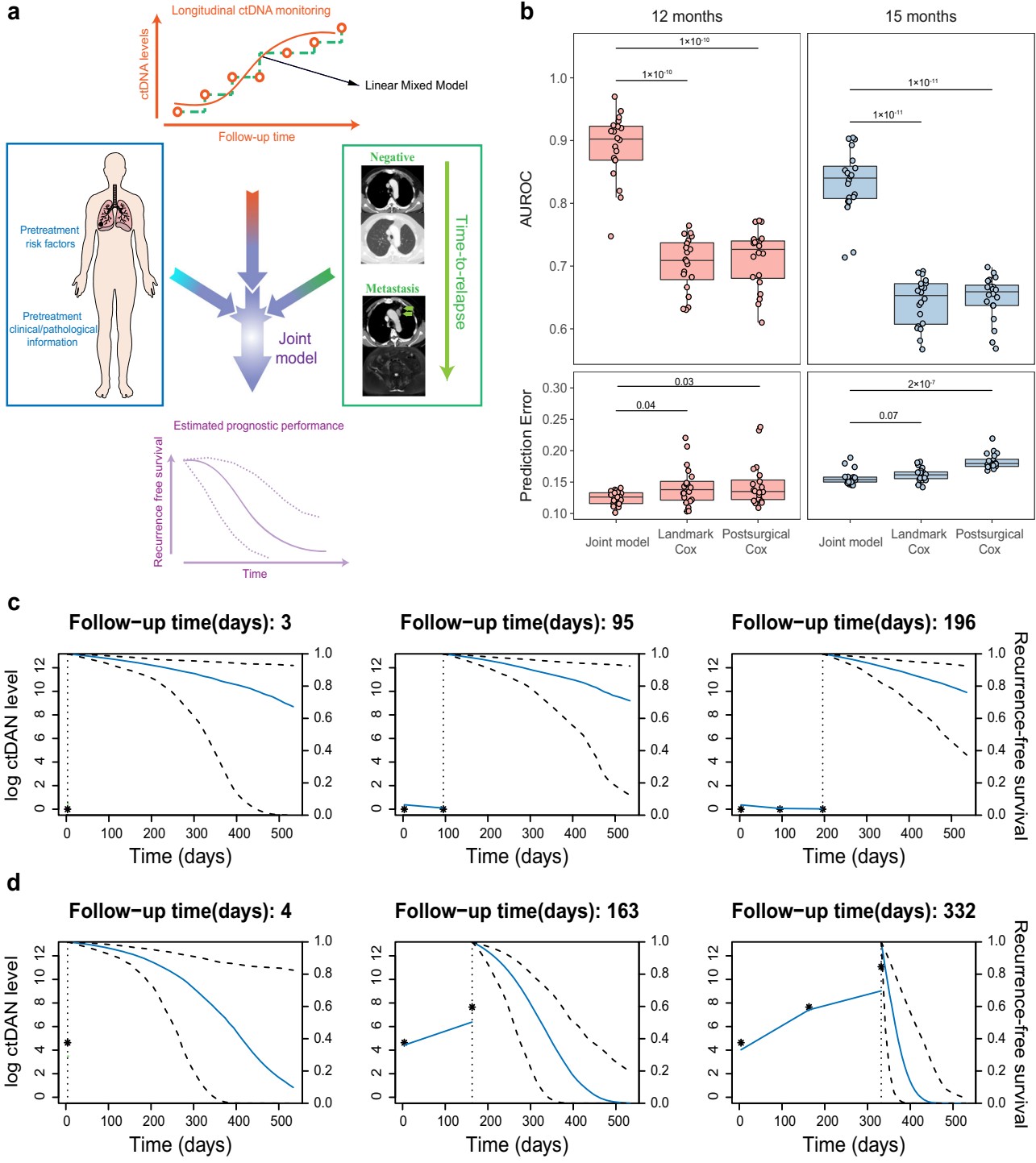

**Fig. 4 Serial ctDNA monitoring for personalized dynamic risk prediction. a** Conception of Joint model. **b** The comparison of model performance between the joint model and cox models (testing datasets). Fivefold cross-validation were repeated for 20 times. The *p* value is calculated using two-sided Wilcoxon signed-rank test. ns: not significant; *$p < 0.05$; **$p < 0.01$; ***$p < 0.001$. Center line, median; box limits, upper and lower quartiles; whiskers, 1.5x interquartile range. AUROC: areas under the receiver-operating characteristics curves. **c, d** Personalized dynamic risk prediction for patients P062 (**c**) and P017 (**d**). From left to right, predictions were calculated accounting for ctDNA that measured previously and were updated when new measure became available. The vertical dotted lines represent the time point of the last ctDNA measurement. To the left of the vertical line is fitted longitudinal trajectory. To the right of the vertical line is the median estimator for recurrence-free probability with 95% pointwise uncertainty band. Log ctDNA level $= \ln(\text{mean VAF} + 10^{-6}) - \ln 10^{-6}$.

they may still be at risk of recurrence and needed to keep under careful observation (Supplementary Fig. 14). By contrast, 79.4% (27/34) of relapsed patients had detected ctDNA during the surveillance, which is similar to two other studies that focused on

surgical patients[29,30]. Without clinical intervention, almost all of the serial samples persisted positive, except for one postsurgical sample (P081) that was collected 3 days after surgery considering a late clearance of ctDNA. A total of seven relapsed patients were

negative for plasma ctDNA before recurrence. One patient (P038) only had brain metastasis upon recurrence. Considering that the blood–brain barrier may potentially prevent the release of its ctDNA into the blood circulation[31] and our ctDNA samples were solely collected from the peripheral blood, it may explain why patient P038 had negative ctDNA detection. Patients P024 and P079 had no detectable ctDNA even at baseline, and the VAF of pretreatment ctDNA in patient P037 was as low as 0.04%, suggesting that the tumors in these patients may not actively release ctDNA. Another reason for a negative ctDNA detection could be the heterogeneity of the primary tumor. That is, the seeding clones of the relapsed tumor might not be included in the sequenced region of the tumor biopsy and the mutated genes of the relapsed tumor were not covered by our testing panel. Also, there was no significant difference in coverage depth and related genes of tumor-specific mutations in relapsed patients where ctDNA was detected compared to those ctDNA that were not detected.

We showed that longitudinal posttreatment ctDNA status could identify NSCLC patients with a higher risk for recurrence and preceded radiological recurrence by a median of 88 days. Although most of the previous NSCLC studies demonstrated that ctDNA detection preceded radiological recurrence, the lead times were not consistent among these studies. The specific clinical characteristics in each study cohort (e.g., pathological stage and treatment strategy) may influence the power of ctDNA preceding radiological recurrence. Abbosh et al. reported a median lead time of 70 days of ctDNA in resected NSCLC in TRACERx study[16] with a three-month repeated blood collection period, which was similar to both the blood collection period (3 months) and median lead time (88 days) in our study, and they updated their data in the year 2020 with the median ctDNA lead time of 151 days[29]. In contrast, Chen et al. conducted another resected NSCLC study and they reported a longer median lead time of 165 days[30]. However, in Chen et al.'s study, 65.4% of patients were stage III or above, compared with that of 49% in our cohort, and their cohort size was relatively small ($n = 26$). Unfortunately, our study was also partially affected by the COVID-19 pandemic. A total of 11 patients in our cohort were delayed on their arranged blood drawn time point, and therefore had their blood samples collected right before or at the time of recurrence. As a result, the lead times in these patients may be largely underestimated. In gastric cancer, ctDNA helped clarify equivocal imaging and CEA findings[13]. We observed similar cases (Fig. 3c) in our study, suggesting that ctDNA can be a great adjunct to radiologic imaging for disease monitor.

Previous studies have demonstrated the potential of MRD detection using only tumor-specific mutations. This approach is mainly limited by tumor heterogeneity. A tumor agnostic approach, which overcomes tumor heterogeneity and can reduce the cost of sequencing, has been proposed. However, its utility is mainly limited by clonal hematopoiesis[32–35], that is, a large fraction of plasma mutations come from clonal hematopoiesis, which lowers the signal-to-noise ratio of the assay. To avoid this shortage, ultradeep sequencing should be also applied to the normal control samples in order to filter out clonal hematopoiesis-related mutations. Another approach using multiplex PCR-based NGS of patient-specific mutations (i.e., 16 mutations selected based on whole-exome sequencing) can track more mutations. Abbosh et al. used a patient-specific anchored-multiplex PCR method which tracked a median of 196 clonal and sub-clonal variants per patient and underlined the capability of ctDNA in both detecting MRD following surgery and defining the clonality of relapsing disease[29]. However, this approach has relatively high costs and is time-consuming, which makes it unsuitable for fast decision-making of clinical applications such as ACT.

ctDNA has been used to guide different treatments. For example, blood tumor mutational burden (bTMB), which was estimated using ctDNA, can be used to stratify NSCLC patients for immunotherapy[36,37]. Moding et al. proposed that personalization of consolidation immune checkpoint inhibitors (ICI) based on the presence of ctDNA after CRT could be a powerful approach for rational therapy selection. Also, Osimertinib showed favorable outcomes in NSCLC patients harboring the *EGFR* T790M mutation, which was also detected from ctDNA[38]. Other therapeutic strategies for lung cancer, such as neoadjuvant chemotherapy or adjuvant immunotherapy, have shown their potential to improve the survival of patients and have prompted substantial interest. The methodologies used in this study, including deep next-generation sequencing and joint modeling, can be easily expanded to neoadjuvant settings in future studies.

In this study, we focused on the potential of ctDNA in ACT treatment after surgery. Taieb et al. have underlined the prognostic value and relation with adjuvant treatment duration of ctDNA in colon cancer[39]. But currently, decision-making for ACT treatment is based on stage and clinical risk factors in lung cancer. A previous meta-analysis of 26 trials of ACT after surgery in NSCLC demonstrated a ~4% absolute survival benefit at 5 years[7], meaning a significant subset of patients enrolled in these trials were cured by surgery and thus may not benefit from adjuvant therapy. In our cohort, although postsurgical ctDNA positivity was significantly associated with worse RFS regardless of ACT treatment, the adjuvant treatment showed a beneficial effect in patients with positive postsurgical ctDNA status. Hence, our study suggests the potential to use ctDNA MRD to stratify a subgroup of ctDNA positive patients who are more likely to benefit from ACT and avoid the overtreatment for ctDNA negative patients. This postsurgical ctDNA negative group could be monitored by ctDNA-based surveillance instead of ACT. However, as the result was based on stage II–III patients, the application on patients who were not recommended for ACT in current clinical settings requires further studies. In addition, although plasma samples were collected prospectively for ctDNA profiling, our analyses were retrospective and patients were not randomly assigned to treatment cohorts. Randomized studies are required to definitively test the clinical utility of ctDNA to guide ACT treatment after surgery.

In this study, we preliminarily demonstrated that joint modeling of longitudinal ctDNA level and time-to-recurrence data can predict the time of recurrence with relatively high accuracy in a cohort of resected NSCLC patients and also illustrated the use of the joint model for individualized dynamic recurrence prediction. Based on the longitudinal measures of ctDNA, the risk of recurrence was calculated and updated when more measurements became available. This approach can guide the personalized assessment frequency and facilitate earlier diagnoses, thus enhancing prognostication and improving the timing to intervention with a disease-modifying agent once available. Further clinical trials need to be done to investigate the effect of the longitudinal outcome on time to event and construct a more accurate relationship between ctDNA change and recurrent progression.

There are potential limitations in our study. We could not collect or test some blood samples based on the schedule due to the COVID-19 pandemic, so the lead time of ctDNA may be underestimated and the overall performance of joint models could not be evaluated and the model may not reach its best performance for personalized dynamic risk prediction. Furthermore, as only internal validation was used to validate the performance of the joint modeling, an independent patient cohort is warranted to further validate the joint model in the future. Although our cohort size is larger than previous studies[15,16], our

analyses of investigating the potential of using postsurgical ctDNA to guide ACT and using joint modeling of longitudinal ctDNA to predict recurrence risk may be limited by the modest sample size of subcohorts.

In summary, we demonstrated that ctDNA serves as a robust biomarker for postsurgical and post-ACT risk stratification and early detection of recurrence in NSCLC. Moreover, our data suggest that postsurgical ctDNA analysis could guide ACT treatment decisions and avoid overtreatment in patients who were less likely to benefit from ACT. We also explored the potential of joint modeling of serial ctDNA and time-to-event data for individualized dynamic risk prediction and demonstrated its superiority over the traditional landmark Cox model. These findings can provide a framework for future clinical trials to investigate the clinical benefits of ctDNA-guided treatment decisions.

## Methods

**Study design.** This was a prospective cohort study of patients with resectable non-small cell lung cancer (NSCLC) enrolled at the Cancer Hospital of Chinese Academy of Medical Sciences from 2018 to 2020. Eligible patients underwent tumor resection with curative intent, followed by adjuvant therapy when indicated by the standard of clinical guidelines. Tumor tissue collected at surgery and pre-treatment peripheral blood samples collected before surgery were used for pre-surgical mutational profiling. Plasma samples were prepared within 2 h of blood collection for ctDNA extraction, while the white blood cells from the buffy coat after plasma preparation were also collected from the same patient at baseline and sequenced as normal controls to identify germline mutations and mutations due to clonal hematopoiesis. The mean sequencing coverage depth of the white blood cells was ~300×. The postsurgical blood samples were collected within 30 days after surgery. Patients were then scheduled to be followed every 3 months with computed tomography (CT) scan and blood collections until recurrences determined by CT scan results. The genetic tests were performed in a centralized clinical testing center (Nanjing Geneseeq Technology Inc., China; Certified to CAP, CLIA and ISO15189) according to protocols reviewed. The study was approved by the Ethics Committee of Cancer Hospital, Chinese Academy of Medical Sciences and Peking Union Medical College. All patients provided oral and written informed consent to participate and publication. This study was registered at Chinese Clinical Trial Registry (ChiCTR) (ChiCTR1900024656; data of registration 20/07/2019).

**Development of lung cancer tracking panel and ATG-seq technology.** A targeted next-generation sequencing panel (Nanjing Geneseeq Technology Inc.) covering 139 critical lung cancer-related genes (Supplementary Table 5) with a total genomic region of 130 kb was used for both tumor tissue and plasma ctDNA specimens. The lung cancer tracking panel was developed based on somatic mutational profiles of more than 8000 Chinese NSCLC tumor specimens and ~500 LUAD and LUSC samples from The Cancer Genome Atlas (TCGA; http://cancergenome.nih.gov) database. Hotspot exons/regions, including all known actionable variants in lung cancer, were included aiming to cover the maximum patients with the minimum panel size for optimal cost-effectiveness. The resulted panel can identify somatic mutations in over 95% of patients in the total database, and over 98% of Chinese lung cancer patients.

To sensitively and specifically detect low-abundance mutations in circulating cell-free DNA (cfDNA), a customized library preparation with bi-barcoding system and ultradeep sequencing approach called Automated Triple Groom Sequencing (ATG-Seq) developed by Nanjing Geneseeq Technology Inc. was applied to cfDNA samples. To minimize the errors from PCR, hybridization, damaging, sequencing, and contamination, and avoid mutations from non-tumor sources in cfDNA, we conducted the following procedures: (i) we sequenced the cfDNA fragment at a depth of ~30,000×, which produced redundant DNA molecules; (ii) mapping positions and a bi-barcode system were used to maximize the representative power of unique DNA molecules; (iii) a duplex assisted decoder system was used to filter mapping and sequencing artifacts;

We also constructed a bioinformatics polishing pipeline by sequencing a pool of plasma samples collected from over 100 healthy donors. Briefly, we performed ATG-seq on plasma samples of healthy individuals to assemble a position- and base substitution-specific background error database based on allele frequency and distinct supporting reads throughout the panel. An alternation was considered as sequencing noise if its allele frequency and distinct supporting reads were not significantly higher than the corresponding background errors in the database. The limit of detection (LOD) on variant allele frequency (VAF) was 0.01%, as tested with ~30,000× deep sequencing of DNA mixtures of two reference human DNA samples (NA19240 and NA18535).

**Library preparation and sequencing.** The plasma was separated by centrifugation at $3000 \times g$ for 10 min. Cell-free DNA (cfDNA) from the plasma was extracted

using QIAmp Circulating Nucleic Acid Kit (Qiagen). FFPE tumor samples were de-paraffinized with xylene, and genomic DNA was extracted using QIAamp DNA FFPE Tissue Kit (Qiagen). Genomic DNA of the white blood cells were extracted using DNeasy Blood & Tissue kit (Qiagen). Purified genomic DNA was qualified by Nanodrop2000 for A260/280 and A260/A230 ratios (Thermo Fisher Scientific). All DNA samples were quantified by Qubit 3.0 using the dsDNA HS Assay Kit (Life Technologies) according to the manufacturer's recommendations.

Sequencing libraries were prepared using KAPA Hyper Prep kit (Roche) with an optimized manufacturer's protocol. In brief, for tumor tissue and normal control samples, 1–2 μg of genomic DNA, which was sheared into ~350 bp fragments using Covaris M220 instrument (Covaris), underwent end-repairing, A-tailing and ligation with indexed sequencing adapters sequentially, followed by size selection using Agencourt AMPure XP beads (Beckman Coulter). For plasma samples, up to 50 ng of cfDNA, underwent end-repairing, A-tailing, ligation with customized adapter containing unique molecular index (UMI), PCR amplifying with primers containing demultiplexing indices sequentially, followed by purification of cfDNA libraries using Agencourt AMPure XP beads (Beckman Coulter). Finally, libraries were amplified by PCR and purified using Agencourt AMPure XP beads.

Different libraries with unique indices were pooled together in desirable ratios for up to 2 μg of total library input. Human cot-1 DNA (Life Technologies) and xGen Universal blocking oligos (Integrated DNA Technologies) were added as blocking reagents. Customized xGen lockdown probes (Integrated DNA Technologies) targeting 139 lung cancer-relevant genes were used for hybridization enrichment. The capture reaction was performed with Dynabeads M-270 (Life Technologies) and xGen Lockdown hybridization and wash kit (Integrated DNA Technologies) according to the manufacturers' protocols. Captured libraries were on-beads PCR amplified with Illumina p5 (5′ AAT GAT ACG GCG ACC ACC GA 3′) and p7 primers (5′ CAA GCA GAA GAC GGC ATA CGA GAT 3′) in KAPA HiFi HotStart ReadyMix (KAPA Biosystems), followed by purification using Agencourt AMPure XP beads. Libraries were quantified by qPCR using KAPA Library Quantification kit (KAPA Biosystems). Library fragment size was determined by Bioanalyzer 2100 (Agilent Technologies). The target-enriched library was then sequenced on HiSeq4000 NGS platforms (Illumina) according to the manufacturer's instructions.

**Mutation calling.** Trimmomatic[40] was used for FASTQ file quality control, leading/trailing low quality (quality reading below 30) or N bases were removed. Qualified reads were then mapped to reference human genome (hg19) using Burrows-Wheeler Aligner[41]. PCR duplicates were removed by Picard (Broad Institute, MA, USA) after local realignment around known indels and base quality recalibration using Genome Analysis Toolkit (GATK 3.4.0). For tissue specimens, Single-nucleotide variations (SNVs) and insertion/deletion were detected using VarScan2[42] with default parameters. Genomic fusions were identified by FACTERA[43] with default parameters. Mutations that were observed in ≥20 cancer cases reported in the COSMIC database were defined as hotspots. a minimum variant allele frequency of 1% or 2% and minimum variant supporting reads of 5 or 6, for hotspot mutations or other mutations, respectively.

For cfDNA samples, single-stranded consensus sequences (SSCS) were generated by collecting all read pairs with the same mapping positions and grouping them into different SSCS families with the same UMI barcode sequences at both ends. Here, we required that a consensus read was supported by at least two reads. After the construction of the SSCS sequence, two SSCS read pairs with transposed UMI barcode sequences and the same mapping position were merged into one DCS, whenever possible. The mean coverage of the single-stranded consensus read was ~6796× (Supplementary Data 3). Hereafter, a local bioinformatics polishing pipeline was used to identify somatic variants in ctDNA after filtering out germline variants using normal control DNA. Mutations identified in the matched tumor DNA, which were supported by a minimum of one unique consensus mutant allele read and passed the polishing criteria were regarded as being present. ctDNA positivity was defined by accessing the presence of one or more mutations identified in the matched tumor sample in ctDNA.

**Joint model construction and evaluation.** To construct the joint model, we combined a linear mixed-effect sub-model to describe the change of ctDNA during the serial measurements and a Cox proportional hazards sub-model for the risk of relapse. The ctDNA variant allele frequency (VAF) at each time point was calculated by averaging the somatic allele frequency for all mutations used for detection calling. Natural cubic splines with two degrees of freedom were used for both the fixed- and random-effects parts of the mixed model based on the investigation of the shapes of the log VAF. In the Cox proportional hazards sub-model, we controlled for T stage and $TP53$ status as baseline covariates (Fig. S4). The joint model was represented as

$$h_i(t) = h_0(t)\exp\{\gamma_1 TP53_i + \gamma_2 Tstage_i + \alpha_1 \eta_i(t)\} \qquad (1)$$

The hazard function was denoted as $h_i(t)$, and $h_0(t)$ was the baseline hazard function. An association parameter $\alpha_1$ linked the two-component sub-models, assuming the hazard at time $t$ was dependent on the true longitudinal trajectory, $\eta_i(t)$, through its value at time $t$. In addition, we compare the performance of current value with two other assumptions of association two sub-models, (i)

current value + current rate of change with hazard function

$$h_i(t) = h_0(t)\exp\{\gamma_1 TP53_i + \gamma_2 Tstage_i + \alpha_1 \eta_i(t) + \alpha_2 \eta_i'(t)\} \quad (2)$$

and (iii) current value + cumulative effect (area under the curve) of ctDNA with hazard function

$$h_i(t) = h_0(t)\exp\left\{\gamma_1 TP53_i + \gamma_2 Tstage_i + \alpha_1 \eta_i(t) + \alpha_3 \int_0^t \eta_i(s)\mathrm{d}s\right\} \quad (3)$$

The best performance was achieved by the joint model that used the current value and cumulative effect (area under curve) as association parameters (Supplementary Fig. 15) and was used as the final joint model.

We adopted a Bayesian approach for model inference and for making predictions using an R package JMbayes[44]. To assess the model predictive performance of the joint model, we examined the discrimination power and calibration. Discrimination power is the ability to separate patients who will relapse early from those who will relapse late or not at all. Calibration measures accurately the model's predictions match overall observed event rates. We estimated the discrimination power and calibration using AUROC and PE, respectively, as proposed by Rizopoulos and calculated them using Jmbayes package. For comparison, the landmark Cox model was fitted using ctDNA detection status and baseline covariates, including T stage and TP53 status. Static Cox model was also fitted using postsurgical ctDNA status and the same baseline covariate. A repeated (20 times) five-fold cross validation was conducted to avoid overestimation of the predictive performance of the markers. Briefly, all patients were randomly split into five subgroups of equal size. The analysis was repeated five times with one subgroup as the test set and the other subgroups together as the training set. Models were fitted with the training set and then applied to the test set. Prediction performance was calculated by averaging the result of these five analyses.

We also constructed the joint model and the landmarking cox model using leave-one-out cross-validation method and evaluated the calibration by reliability diagrams using the Hosmer-Lemeshow test[27], a statistical test for goodness of fit. The data are divided into a number of groups, where each group contains approximately the same number of patients. The observed and expected number of cases in each group is calculated and a Chi-squared statistic is calculated (C-statistic). To account for censoring, we calculated the observed event probability by the Kaplan–Meier method.

**Statistical analyses**. Recurrence-free survival (RFS) was measured from the date of surgery to the verified first radiographic recurrence (local or distant). Cox proportional hazards regression analysis and Kaplan–Meier estimate were used to assess the association of ctDNA and clinical variables. Multivariate analysis was performed with clinical variables that were statistically significant in univariate analysis. Wilcoxon signed-rank test was used to test the significance of lead time from ctDNA detection to radiographic recurrence and AUROC and PE of joint model and cox models. All $P$ values were based on two-sided testing, and differences were considered significant at $P < 0.05$. Statistical analysis was performed using R software, version 4.0.2.

**Reporting summary**. Further information on research design is available in the Nature Research Reporting Summary linked to this article.

## Data availability

All raw targeted DNA-sequencing data have been deposited in the National Genomics Data Center (NGDC) under the accession code HRA001346. The deposited and publicly available data are compliant with the regulations of the Ministry of Science and Technology of the People's Republic of China. The raw sequencing data contain information unique to individuals and are available under controlled access. Access to the data can be requested by completing the application form via GSA-Human System and is granted by the corresponding Data Access Committee. Additional guidance can be found at the GSA-Human System website [https://ngdc.cncb.ac.cn/gsa-human/document/GSA-Human_Request_Guide_for_Users_us.pdf]. Data used for survival analysis and joint model construction and evaluation are publicly available at https://github.com/cancer-oncogenomics/ctDNA-dynamic-prediction-lung-cancer. All specific mutation genomic locations and allele frequencies are available in Supplementary Data 2. Source data are provided with this paper.

## Code availability

All analyses were performed using R version 4.0.2. R package survival (version 3.2-10) was used for survival analysis. R package JMbayes (version 0.8-85) was used for the construction and evaluation of joint models and cox models. Reference scripts to reproduce the results of this study is available at https://github.com/cancer-oncogenomics/ctDNA-dynamic-prediction-lung-cancer.

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

## Acknowledgements
We appreciate the support and participation of the physicians and patients in this study.

## Author contributions
B.Q. and S.G. conceived the study. B.Q., W.G., H.B., Y.S., F.T., Q.X. and S.G. provided project management and supervision. B.Q., W.G., F.Z., F.L., Y.J., Y.P. and F.T. provided or facilitated the accrual of patient samples, pathology, and/or clinical data. W.G., X.C. and H.B. performed bioinformatics and genomic analyses. B.Q., W.G., F.Z., F.L., Y.J. and Y.P. performed statistical analyses. W.G. and X.C. wrote the original draft, with input from all authors. B.Q., Y.X., H.B., S.G. and J.H. review and editing the manuscript.

## Competing interests
The following authors are employees of Nanjing Geneseeq Technology Inc. (Xiaoxi Chen, Hua Bao, Yang Xu, and Yang Shao). All remaining authors declare no competing interests.
