## [Peer Review File · Nature Communications]

Reviewers' Comments:

Reviewer #1:

Remarks to the Author:

The authors present a study on the utility of (serial) ctDNA measurement in prediction of outcomes in lung cancer patients. Firstly, the authors highlight that pre/post-treatment ctDNA levels in lung cancer patients can predict recurrence and inform use of adjuvant therapy. However, these findings are not very surprising given the existing literature. Next, the authors explore the benefit of serial ctDNA measurements as input to joint/longitudinal modelling of patient outcomes. This is a very interesting idea. However, the approach is only vaguely described and the model validation is somewhat lacking (see specific comments below). Furthermore, the manuscript suffers from the fact that the cohort is mixture of Lung Adenocarcinoma and Squamous Cell Carcinoma patients, which tend to have different baseline outcomes and probability of ctDNA detection (see comment below). This (and the missing figure legends?) makes it very hard to interpret the results of the paper. Below are my specific major and minor comments.

Major

The joint/longitudinal model is poorly described and validated

A key innovation of the paper is the evaluation of a longitudinal/joint model of serial ctDNA samples. However, this model is only superficially described at the end of the paper, and the validation is lacking.

- * The manuscript should include a main figure that explains the joint model concept and how it differs from a standard/landmark Cox model.

- * The authors must show both training and test set errors across all 20 splits training/test set samples. Ideally, the final model should be evaluated on a completely independent/withheld set of patients.

- * The simple Cox model (landmark) appear to have comparable or lower prediction error than the joint model. What's the advantage of the joint model then?

- * The splits for training / test sets and the data and code used to fit and evaluate the models must be included.

- * The model training/evaluation should also be evaluated separately for squamous and adeno tumor types (see below).

Figure legends??

I looked through both the manuscript and supplemental document and couldn't find any figure legends (I am sorry if I somehow missed them). There were many figures I couldn't interpret as a result.

Survival analysis does not account for tumor type

The authors use a mixed cohort of early stage lung cancer patients comprising different tumor types. Patients with distinct tumor types (eg Lung Adeno vs. Squamous) could have substantially different expected/baseline treatment outcomes. Furthermore, the authors note that Lung Adenocarcinoma tumors have noticeably lower ctDNA detection frequency. Overall, these factors could confound the different types of survival analysis conducted throughout the paper. The authors must ensure that the main conclusions in this paper is not affected by the mixing of tumor types in their cohort, including the survival analysis presented in Fig. 2a,b,c,d, 3b, 4a,b,c.

Potential of postsurgical ctDNA to guide ACT

The analysis presented in this section is confusing and potentially misleading. First, the authors compare outcome for patients treated with and without ACT among all ctDNA-positive patients. Next, they do same for ctDNA-negative patients. To support the conclusion that ctDNA can guide ACT, it would seem more logical if the authors show that ctDNA-status (presence/absence) can stratify ACT-treated patients into low and high-risk groups. This would demonstrate that ctDNA could guide ACT treatment beyond existing stratification approaches used in the clinic. Again, this analysis should take into account the different tumor types in their mixed cohort. Finally, the labels in Fig 2c are not logical ('PoS.ctDNA-Neg Act-No'), please clean up this figure.

Data sharing and reproducibility

It is not clear if the authors are sharing code and data ensuring reproducible research. The authors should share all code and data that reproduces figures and key survival analysis in this manuscript. The raw targeted sequencing data for both tumor and liquid biopsies should also be shared.

Variant calling without matched normals

In the mutation calling part (for both tumor and plasma samples), it seems the author did not sequence the matched buffy coat? It has been demonstrated the sequencing of matched cfDNA–white blood cell is important for accurate variant interpretation in the presence of Somatic Hematopoiesis (Razavi et al., Nature Medicine, 2019). The authors must discuss the impact of this omission.

Personalised models of recurrence risk

Fig. 4c showed two representative patients to suggest the method accurately estimate personalized recurrence risks. The author should show the concordance between ctDNA and recurrence for all available patients, not only the selected two patients. A robust method should generalized well across most patients.

Minor

* The VAFs from mutation calling (tumor tissue and cfDNA) as well as specific locations corresponding to each gene in the panel should be fully provided.

* In the Study Design, the author say the follow-up is 3 months for CT-scan and blood assay. However, Fig. 3a shows the time intervals between time points are not uniform. Please clarify.

* The top rows of Table 1 are damaged.

* Reference is needed for this statement: "Although the imaging technique offers an assessment of the tumor burden, it cannot reliably detect minimal residual disease (MRD)."

* References are needed for this statement: "In recent years, joint modelling of longitudinal and time-to-event data becomes more and more popular in clinical trial studies."

Reviewer #2:

Remarks to the Author:

Guo et al. presents a study on potential implications of ctDNA findings for patients with resectable NSCLC. The study tackles important clinical questions and the optimal study setting and encouraging results provides very interesting insights. The data is nicely presented, however particularly the discrepancies between some of the ctDNA findings and the clinical outcome and the limitations of the applied methodology, needs to be much more thoroughly addressed.

General:

- I think the methodology needs to be presented with vastly more details, as although the results are quite encouraging, it is troublesome that so many patients with metastasis do not have detectable ctDNA (fig 3a). However, this could also simply reflect interpretation of too few mutation targets per patient/sample, but this needs to be discussed.

And that a handful of patients have no recurrence despite having detectable ctDNA.

- The joint model approach provides an interesting angle to the data. However, I think it is lacking details in the figure legends. In addition, as I understand it, it optimally should be based on multiple plasma samples and I wonder if, ideally, ACT should be administered when ctDNA is reliably detected just once and several ctDNA measurements for tracking the progression of VAFs, should hopefully not be necessary.

- Overall, I believe the findings contribute with valuable information, however the referencing of

previous work is fairly lackluster. Several recent studies have e.g. investigated ctDNA dynamics during treatment and identified interesting results doing so for NSCLC. Increased referencing and relation to previous ctDNA based NSCLC studies would improve the manuscript.

Methodology:

- How are ctDNA mutation calls defined, i.e. what is the required level of support for a mutation to be called?
- Is a bi-barcode system equivalent to unique molecular identifiers on both strands as presented by Kennedy et al, Nature Protocols 2014?
- If so, what is the resulting unique sequencing depth?
- How large was the pool of samples from healthy donors and how is it leveraged in the reduction of errors?
- Based on all of the above, what are the expectations or preferable what are the experimental/in silico limit of detection for the analysis protocol employed here? This is crucial to address as it currently remains unclear if the findings with a lack of correlation between ctDNA status and patient outcome is of biological or technical nature.
- Furthermore, how is the tumor guidance performed? How frequently do you see other mutations in the panel occurring? And what are your thoughts about interpretation of those, as they might also represent mutations from tumor clones not sampled in the tumor samples?
- And most importantly, would it be possible to use the strategy as a tumor agnostic approach so no tumor NGS data is necessary?

Minor:

- What was the volume of plasma used for analysis? What was the median extracted amount of cfDNA? Did you assess purification efficiency in any way?
- Please provide arrows and/or explanations for the CT images.
- Please elaborate on Fig S5

Response to Reviewers' comments (reviewer comments in italics):

Reviewer #1

The authors present a study on the utility of (serial) ctDNA measurement in prediction of outcomes in lung cancer patients. Firstly, the authors highlight that pre/post-treatment ctDNA levels in lung cancer patients can predict recurrence and inform use of adjuvant therapy. However, these findings are not very surprising given the existing literature. Next, the authors explore the benefit of serial ctDNA measurements as input to joint/longitudinal modelling of patient outcomes. This is a very interesting idea. However, the approach is only vaguely described and the model validation is somewhat lacking (see specific comments below). Furthermore, the manuscript suffers from the fact that the cohort is mixture of Lung Adenocarcinoma and Squamous Cell Carcinoma patients, which tend to have different baseline outcomes and probability of ctDNA detection (see comment below). This (and the missing figure legends?) makes it very hard to interpret the results of the paper. Below are my specific major and minor comments.

Reply:

We thank the Reviewer for summarizing our work, careful consideration of our results, and perceptive suggestions. We added more details to explain the joint model and performed additional analyses to demonstrate the advantage of joint model compared to the traditional cox regression. In addition, we analyzed the result of adenocarcinoma and squamous cell carcinoma patients separately.

Major

The joint/longitudinal model is poorly described and validated

A key innovation of the paper is the evaluation of a longitudinal/joint model of serial ctDNA samples. However, this model is only superficially described at the end of the paper, and the validation is lacking.

** The manuscript should include a main figure that explains the joint model concept and how it differs from a standard/landmark Cox model.*

Reply:

We appreciate the comments from the Reviewer. In the revised manuscript, we added more details about the joint model and the validation method in the Methods and Results sections. A main schematic diagram (Fig. 4a) explaining the joint model concept was also added. Specifically, the joint modeling comprises two linked submodels, one for the longitudinal process (a linear mixed submodel of repeated measurements of ctDNA) and one for the time-to-relapse data (a Cox submodel with baseline covariates). It exploits the full information of collected data during follow-up up to landmark time point, and takes into account the sporadic measurement error of the longitudinal data. In contrast, the traditional landmark Cox model for dynamic risk prediction applies survival model based solely on the last observed value of the biomarker at the landmark time and obtains survival probabilities from a Cox model fitting to the patients who are still at risk at the time point of interest.

Changes in the revised version (clean version): Lines 209-229, 436-475

* *The authors must show both training and test set errors across all 20 splits training/test set samples. Ideally, the final model should be evaluated on a completely independent/withheld set of patients.*

Reply:

We thank the Reviewer for the suggestion and agree that an independent set of patients as external validation would be the ideal way to evaluate the model. Unfortunately, there was no external dataset available for validation in this cohort. Instead, we performed repeated ($n=20$) 5-fold cross-validation to validate our results and estimated the model accuracy. The training and validation samples were completely independent/withheld. These results were shown in the revised Fig. 4b (as shown below), which includes the AUC and prediction error results of the 20 testing sets, and the newly added Supplementary Fig 10 (as shown below), which includes the AUC and prediction error results of the 20 training sets. Results of each split were provided in Source Data file. In the future, we will recruit an independent patient cohort to validate the joint model.

Figure 4b The comparison of model performance of testing dataset between the joint model and cox models. The comparison of model performance between the joint model and cox models (testing datasets). The p value is calculated using Wilcoxon signed rank test. ns: not significant; *: $p < 0.05$; **: $p < 0.01$; ***: $p < 0.001$. Center line, median; box limits, upper and lower quartiles; whiskers, 1.5x interquartile range.

Supplementary Fig. 10 The comparison of model performance of training datasets between the joint model and cox models. The p value is calculated using Wilcoxon signed rank test. ns: not significant; *: $p < 0.05$; **: $p < 0.01$; ***: $p < 0.001$. Center line, median; box limits, upper and lower quartiles; whiskers, 1.5x interquartile range.

* The simple Cox model (landmark) appear to have comparable or lower prediction error than the joint model. Whats the advantage of the joint model then?

Reply:

We appreciate the comments from the Reviewer and understand his/her concerns. In the revised manuscript, we provided more detailed comparison of different models to demonstrate the advantage of the joint model.

To better evaluate the model performance, we used a Monte Carlo approach instead of a first-order estimator in the ‘jmbayes’ package setting (Simulate=True; See the reference scripts) to estimate the survival probability when calculating the AUROC and prediction error. The Monte Carlo approach is more accurate but more computationally intensive¹. The updated results

were shown in the above revised Fig. 4b, the newly added above Supplementary Fig. 10, and the newly added Supplementary Table 6 (as shown below).

Firstly, the joint model achieved a significantly higher discrimination power with a significant higher AUROC when compared with the landmarking Cox models (Wilcoxon rank test; $P < 0.0001$ at both 12 months and 15 months; revised Fig. 4b upper panels), which indicated that the joint model has superior ability to separate early relapsed patients from late relapsed/non-relapsed patients than the landmarking Cox model.

Secondly, the updated prediction error of joint model was lower than landmarking Cox model at 12 months (0.12 vs. 0.14) and similar at 15 months (0.16 vs. 0.16) (Fig. 4b bottom panels; Supplementary Fig. 10; Supplementary Table 6). To estimate prediction error, we also constructed the joint model and landmarking Cox model using the leave-one-out cross-validation method (more computationally intensive) and evaluated the calibration by reliability diagrams using Hosmer-Lemeshow (H-L) C-statistics². H-L statistics and P values results were calculated and shown in the newly added Supplementary Fig. 11 (as shown below). In both 12-month and 15-month results, the agreement between the predicted and observed survival probabilities was higher in the joint model (large P values and smaller C-statistic values) than the Cox regression.

Supplementary Table 6. Performance of joint models and Cox models

	Prediction time (months)	type	JM (value)	JM (value+slope)	JM (value+cumulative)	Cox (Landmark)	Cox (Postsurgical)
AU C	12	training	0.89	0.86	0.90	0.73	0.78
		testing	0.84	0.83	0.89	0.70	0.73
	15	training	0.80	0.83	0.83	0.68	0.69
		testing	0.77	0.77	0.83	0.64	0.67
PE	12	training	0.13	0.12	0.11	0.11	0.19
		testing	0.14	0.13	0.12	0.14	0.16
	15	training	0.17	0.16	0.15	0.14	0.22
		testing	0.18	0.17	0.16	0.16	0.20

Supplementary Fig. 11 Reliable diagrams of the final joint model and landmarking cox model. Reliability diagrams of 12-month (a) and 15-month (b) estimates produced by joint model and landmarking cox model grouped for the Hosmer-Lemeshow (H-L) C-statistics. Data points of estimates produced by the models and their actual binary outcomes are plotted to show the distribution of the actual data. The number of patients within each bin is the same. Dashed vertical lines indicates 95% CI.

Changes in the revised version: Lines 230-244

** The splits for training / test sets and the data and code used to fit and evaluate the models must be included.*

Reply:

We thank the Reviewer for the suggestion. The splits for training/test sets as well as reference scripts and data for model construction and evaluation were uploaded to <https://github.com/cancer-oncogenomics/ctDNA-dynamic-prediction-lung-cancer>. We also added ‘Data availability’ and ‘Code availability’ sections in the revised manuscript.

Changes in the revised version: Lines 487-498

** The model training/evaluation should also be evaluated separately for squamous and adeno tumor types (see below).*

Reply:

We appreciate the comments from the Reviewer and understand his/her concerns. In the revised manuscript, we added the evaluation analysis of the model performance using either AD

or SqCC patients. The newly added Supplementary Fig. 12 (as shown below) showed that the joint model had a similar performance when constructing the model using either AD patients or all patients, compared with the inferior performance of the corresponding Cox models who failed to distinguish patients with high or low risk ($AUC < 0.5$). However, for SqCC patients, we failed to construct and evaluate the model due to the small sample size of this sub-cohort of patients.

Changes in the revised version: Lines 242-244

Supplementary Fig. 12 The comparison of model performance between the joint model and cox models using only AD patients. a Testing dataset. **b** Training dataset. Time-dependent areas under the receiver-operating characteristics curves (AUROC) and prediction error (PE) represent discrimination power and calibration of the models. The p value is calculated using Wilcoxon signed rank test. ns, not significant; *, $p < 0.05$; **, $p < 0.01$; ***, $p < 0.001$. Center line, median; box limits, upper and lower quartiles; whiskers, 1.5x interquartile range.

Figure legends are missing??

I looked thought both the manuscript and supplemental document and couldn't find any figure legends (I am sorry if I somehow missed them). There were many figures I couldn't interpret as a result.

Reply:

We apologize for this confusion in our previous submission, as we included the Figure Legends as a separate file named 'Related Manuscript File'. In the revised manuscript, we have moved the Figure Legends to the end of the main text file.

Changes in the revised version: Lines 657-689

Survival analysis does not account for tumor type

The authors use a mixed cohort of early stage lung cancer patients comprising different tumor types. Patients with distinct tumor types (eg Lung Adeno vs. Squamous) could have substantially different expected/baseline treatment outcomes. Furthermore, the authors note that Lung Adenocarcinoma tumors have noticeably lower ctDNA detection frequency. Overall, these factors could confound the different types of survival analysis conducted throughout the paper. The authors must ensure that the main conclusions in this paper is not affected by the mixing of tumor types in their cohort, including the survival analysis presented in Fig. 2a,b,c,d, 3b, 4a,b,c.

Reply:

We appreciate the comments from the Reviewer. Although pretreatment ctDNA detection rate showed a significant difference in different tumor types, there was no difference in postsurgical detection rate (AD vs SqCC, 19.2% vs 19.4%; Fisher's exact test; P=1). One of the main conclusions of our study was that postsurgical, post-ACT and longitudinal ctDNA positivity were significantly associated with worse RFS. To investigate whether the tumor histology subtypes would affect this conclusion, we controlled for tumor types in the multivariate Cox proportional hazard model and found that ctDNA status remained statistically significant (the revised Supplementary Table 3-5). Next, we performed survival analysis in patients with adenocarcinoma (AD) or squamous cell carcinoma (SqCC) (newly added supplementary Fig. 5, as shown below), based on their postsurgical, post-ACT and longitudinal ctDNA detection. Consistent with the results in Fig. 2, postsurgical and longitudinal ctDNA-positive status was associated with worse RFS for both AD and SqCC, although the result for post-ACT ctDNA status was not significant, which is likely due to the small sample size after stratifying patients based on the tumor types.

In addition, we tested the ability of ctDNA to detect recurrence preceding radiographic recurrence in solely AD or SqCC patients. Similar to the results in Figure 3b, ctDNA was detected prior to radiographic recurrence with a median of 20 days in AD patients (Wilcoxon signed rank test; P < 0.05; N=14) and a median of 93 days in SqCC patients (Wilcoxon signed rank test; P < 0.05; N=9). In addition, we evaluated the results of our joint modeling in AD or SqCC patients separately (as shown above) and reached the same conclusion as the combined analysis.

Changes in the revised version: Lines 122-125, 129-131, 145-147, 184

Supplementary Fig. 5 ctDNA positivity was related to patient prognosis in both adenocarcinoma (AD) and squamous cell carcinoma (SqCC) patients. **a-b** Kaplan-Meier curve of RFS stratified by postsurgical ctDNA status in AD patients (**a**) and SqCC patients (**b**). **c-d** Kaplan-Meier curve of RFS stratified by post-ACT ctDNA status in AD patients (**c**) and SqCC patients (**d**). **e-f** Kaplan-Meier curve of RFS stratified by longitudinal ctDNA status in AD patients (**e**) and SqCC patients (**f**).

Potential of postsurgical ctDNA to guide ACT

The analysis presented in this section is confusing and potentially misleading. First, the authors compare outcome for patients treated with and without ACT among all ctDNA-positive patients. Next, they do same for ctDNA-negative patients. To support the conclusion that ctDNA can guide ACT, it would seem more logical if the authors show that ctDNA-status (presence/absence) can stratify ACT-treated patients into low and high-risk groups. This would demonstrate that ctDNA could guide ACT treatment beyond existing stratification approaches used in the clinic. Again, this analysis should take into account the different tumor types in their mixed cohort.

Reply:

We apologize for the confusion and thank the Reviewer for his/her comments and suggestions. We have revised the analysis in the revised manuscript accordingly.

ACT after surgery could potentially eliminate MRD and improve survival, and is recommended for patients clinically diagnosed as high-risk population of recurrence such as stage II and III NSCLC patients. However, a significant proportion of these patients still relapsed. We therefore hypothesized that ctDNA-positive subgroup of patients might benefit from ACT, whereas ctDNA-negative but clinically diagnosed high-risk patients may not benefit from ACT. We stratified all the stage II-III patients, who should be recommended for ACT treatment post-surgery, based on whether they received ACT treatment or whether they had detectable postsurgical ctDNA (Fig. 2c). Consistent with our hypothesis, ctDNA-positive patients had a significant higher risk of recurrency compared to ctDNA-negative patients within either ACT group ($p < 0.05$) or non-ACT group ($p < 0.05$). ctDNA-negative patients have similar low risk of relapsing, independent of whether or not ACT was administered ($p = 0.46$). In contrast, ctDNA-positive patients who treated with ACT had a significantly improved RFS than ctDNA-positive patients without ACT ($p < 0.05$). Therefore, postsurgical ctDNA status could potentially stratify these stage II-III NSCLC patients into two groups, the ctDNA-positive patients who could more likely benefit from ACT treatment, and the ctDNA-negative group where ACT seems to be unnecessary with a minimal improvement in reducing their relapse risk.

We also performed the same analysis in SqCC and AD patients separately to see whether the conclusion is valid across different tumor types. As shown in the newly added Supplementary Fig. 8 (as shown below), ACT treatment was significantly related to better RFS in ctDNA-positive SqCC patients and had trend of better RFS in ctDNA-positive AD patients.

Changes in the revised version: Lines 151-176, 308-311

Supplementary Figure 8 Kaplan-Meier curve of RFS in stage II-III patients stratified by both ACT treatment and postsurgical ctDNA status in AD (a) and SqCC (b) patients

Finally, the labels in Fig 2c are not logical ('PoS.ctDNA-Neg Act-No'), please clean up this figure.

Reply:

Based on the Reviewer's suggestion, we revised the Fig. 2c and re-labelled it to avoid confusions.

Figure 2c Kaplan-Meier curve of RFS in stage II-III patients stratified by both ACT treatment and postsurgical ctDNA status.

Data sharing and reproducibility

It is not clear if the authors are sharing code and data ensuring reproducible research. The authors should share all code and data that reproduces figures and key survival analysis in this manuscript. The raw targeted sequencing data for both tumor and liquid biopsies should also be shared.

Reply:

We thank the Reviewer for his/her comments and suggestions. We added the data that can reproduce Figures in the source data file. Code for model construction and evaluation and survival analysis are available at <https://github.com/cancer-oncogenomics/ctDNA-dynamic-prediction-lung-cancer>. We also provided the specific mutations identified in both tumors and liquid biopsy samples with their allele frequencies in the Supplementary Table 2. However, we currently cannot make the raw targeted sequencing data publicly available due to the regulation of the Human Genetic Resources Administration of China (HGRAC). Request for accessing the raw sequencing data can be sent directly to the corresponding author if the manuscript is accepted for publication with proper HGRAC approval.

Variant calling without matched normals

In the mutation calling part (for both tumor and plasma samples), it seems the author did not sequence the matched buffy coat? It has been demonstrated the sequencing of matched cfDNA–white blood cell is important for accurate variant interpretation in the presence of Somatic Hematopoiesis (Razavi et al., Nature Medicine, 2019). The authors must discuss the impact of this omission.

Reply:

We apologize for the confusion due to the lack of details in the Methods of our previous submission. We did sequence the white blood cells as normal controls to identify germline

mutations and mutations of clonal hematopoiesis. In the revised manuscript, we have added the statement that ‘white blood cells from the buffy coat after plasma preparation were also collected and sequenced as normal controls to identify germline mutations and mutations due to clonal hematopoiesis’ in the Methods section.

Changes in the revised version: Lines 343-345

Personalised models of recurrence risk

Fig. 4c showed two representative patients to suggest the method accurately estimate personalized recurrence risks. The author should show the concordance between ctDNA and recurrence for all available patients, not only the selected two patients. A robust method should generalized well across most patients.

Reply:

We thank the Reviewer for the suggestion. Joint model can discriminate early relapsed patients from late relapsed/non-relapsed patients at both 12 months (AUROC=0.89) and 15 months (AUROC=0.83), which indicated the superior overall concordance between ctDNA and recurrence. Also, we agree that personalized recurrence prediction results of most patients are needed for a robust method. In the revised manuscript, we added all other personalized recurrence results for patients with two or more blood collections in the newly added Supplementary Figure 13.

Minor

** The VAFs from mutation calling (tumor tissue and cfDNA) as well as specific locations corresponding to each gene in the panel should be fully provided.*

Reply:

We thank the Reviewer for the suggestion. In the revised manuscript, we have provided the information in the Supplementary Table 2.

** In the Study Design, the author say the follow-up is 3 months for CT-scan and blood assay. However, Fig. 3a shows the time intervals between time points are not uniform. Please clarify.*

Reply:

As mentioned by the Reviewer, we designed our study to perform the CT-scan and collect the plasma samples every 3 months; however, due to the COVID-19 pandemic, some samples cannot be collected or tested as planned. To avoid the misunderstanding, in the revised manuscript we re-stated the study design as “Patients were then scheduled to be followed every 3 months with computed tomography scan and blood collection”. In addition, we added a limitation section in the discussion part to clarify this point.

Changes in the revised version: Lines 322-325, 346-347

** The top rows of Table 1 are damaged.*

Reply:

We apologize for this inconvenience. In the revised manuscript, we have fixed this problem and re-uploaded the Table 1.

** Reference is needed for this statement: “Although the imaging technique offers an assessment of the tumor burden, it cannot reliably detect minimal residual disease (MRD).”*

Reply:

We thank the Reviewer for the suggestion. We add Chao *et al.*³ and Huang *et al.*⁴ as references in revised manuscript.

** References are needed for this statement: “In recent years, joint modelling of longitudinal and time-to-event data becomes more and more popular in clinical trial studies.”*

Reply:

We thank the Reviewer for the suggestion. We add Ibrahim *et al.*⁵ and Li *et al.*⁶ as references in revised manuscript.

Reviewer #2

Guo et al. presents a study on potential implications of ctDNA findings for patients with resectable NSCLC. The study tackles important clinical questions and the optimal study setting and encouraging results provides very interesting insights. The data is nicely presented, however particularly the discrepancies between some of the ctDNA findings and the clinical outcome and the limitations of the applied methodology, needs to be much more thoroughly addressed.

General:

- I think the methodology needs to be presented with vastly more details, as although the results are quite encouraging, it is troublesome that so many patients with metastasis do not have detectable ctDNA (fig 3a). However, this could also simply reflect interpretation of too few mutation targets per patient/sample, but this needs to be discussed.

And that a handful of patients have no recurrence despite having detectable ctDNA.

Reply:

We thank the Reviewer for evaluating our manuscript and providing useful comments. We agree with the Reviewer that the sensitivity of metastasis detection by ctDNA is mainly limited by the too few mutation targets per patient. We only tracked mutations detected in the matched tumor samples and it could be limited by heterogeneity of the tumor. Nonetheless, by tracking only mutations in the match tumors, we achieved a sensitivity of 82.1% in patients with pretreatment ctDNA shedding and 60% in patients without pretreatment ctDNA shedding, which is similar to two other studies that focused on surgical patients^{7,8}. Another approach using multiplex PCR-based NGS of patient-specific mutations (i.e. 16 mutations selected based on whole-exome sequencing) can track more mutations compared to panel sequencing. However, this approach has relatively high cost and is time-consuming, which is not suitable for fast

decision in clinical applications such as ACT decision. We added these comments in the discussion section.

As for patients who had no recurrence despite having detectable ctDNA, three of them had positive ctDNA before or during ACT. The other two had positive ctDNA at last several time points which indicated that they may still be at risk of recurrence and needed to keep under careful observation.

Changes in the revised version (clean version): Lines 264-275, 281-291

- The joint model approach provides an interesting angle to the data. However, I think it is lacking details in the figure legends. In addition, as I understand it, it optimally should be based on multiple plasma samples and I wonder if, ideally, ACT should be administered when ctDNA is reliably detected just once and several ctDNA measurements for tracking the progression of VAFs, should hopefully not be necessary.

Reply:

We thank the reviewer for the comments. In the revised manuscript, we added more detailed about the joint model in both the figure legends and the method section. Also, we agree with the Reviewer that a reliable one-time ctDNA detection, such as postsurgical detection, can help guide ACT decision. However, we think serial measurements of ctDNA are still necessary because it can help monitor disease recurrence and is a great adjunct for routine clinical imaging. In addition, the joint model approach using serial ctDNA measurements can better predict recurrence and may help guide the post-recurrence treatment.

Changes in the revised version: Lines 418-476, 679-689

- Overall, I believe the findings contribute with valuable information, however the referencing of previous work is fairly lackluster. Several recent studies have e.g. investigated ctDNA dynamics during treatment and identified interesting results doing so for NSCLC. Increased referencing and relation to previous ctDNA based NSCLC studies would improve the manuscript.

Reply:

We thank the Reviewer for the suggestion. We added more discussions about recent studies of ctDNA dynamics in NSCLC and cited the corresponding references in the revised manuscript.

Changes in the revised version: Lines 292-298

Methodology:

- How are ctDNA mutation calls defined, i.e. what is the required level of support for a mutation to be called?

Reply:

For mutations identified in the matched tumor DNA, a minimum of one consensus read in carrying the same variant in the ctDNA were regarded as being present. We constructed a bioinformatics polishing pipeline by sequencing a pool of plasma samples collected from more

than 100 healthy donors. Briefly, we performed ATG-seq on plasma samples of fifty healthy individuals to assemble a position- and base substitution–specific background error database based on allele frequency and distinct supporting reads throughout the panel. An alternation was considered as sequencing noise if its allele frequency and distinct supporting reads were not significantly higher than the corresponding background errors in the database. ctDNA positivity was defined by accessing the presence of one or more mutations identified in the match tumor samples. The above information has been added to the method section of the revised manuscript.

Changes in the revised version: Lines 418-433

- *Is a bi-barcode system equivalent to unique molecular identifiers on both strands as presented by Kennedy et al, Nature Protocols 2014?*
- *If so, what is the resulting unique sequencing depth?*

Reply:

The bi-barcode system we used in this study is similar to the unique molecular identifiers on both strands of Kennedy *et al.* with a differently designed barcode length. The median of mean unique sequencing depth of all plasma samples was 6776.6×

Changes in the revised version: Lines 364-373

- *How large was the pool of samples from healthy donors and how is it leveraged in the reduction of errors?*

Reply:

We thank the reviewer for this comment. We constructed a bioinformatics polishing pipeline by sequencing a pool of plasma samples collected from more than 100 healthy donors and described this above in the ‘ctDNA mutation calls definition’ question. All of the information was added to the method section of the revised manuscript.

Changes in the revised version: Lines 374-379

- *Based on all of the above, what are the expectations or preferable what are the experimental/in silico limit of detection for the analysis protocol employed here? This is crucial to address as it currently remains unclear if the findings with a lack of correlation between ctDNA status and patient outcome is of biological or technical nature.*

Reply:

We thank the reviewer for this comment. In this study, we only tracking tumor-informed mutations and a minimum of one consensus read in carrying the same variant in the ctDNA were regarded as being present. Thus, we estimated the limit of detection when tracking known mutations using mixed reference cell lines⁹. We mixed cDNA from two reference cell lines (NA19240 and NA18535) and created a series of DNA mixtures of lowest VAF at 0.5%, 0.25%, 0.1%, 0.05%, 0.01%, and 0% (negative control samples). We found that ATG-Seq could detect known mutations down to 0.01% with >99% specificity in negative control samples. Also, in this study, ctDNA positivity after surgical resection, after ACT treatment and during the surveillance

was all correlated with poor patients' outcomes, indicating that ctDNA detection status was well correlated with patient outcome.

Changes in the revised version: Lines 379-381

- Furthermore, how is the tumor guidance performed? How frequently do you see other mutations in the panel occurring? And what are your thoughts about interpretation of those, as they might also represent mutations from tumor clones not sampled in the tumor samples?

Reply:

We thank the reviewer for the comment. Overall, by using the tumor guidance approach to track ctDNA in relapsed patients, we achieved a sensitivity of 82.1% in patients with pretreatment ctDNA shedding and 60% in patients without pretreatment ctDNA shedding.

Among all the plasma samples, 58.0% (258/445) had mutations that were not detected in the match tumors, 61.2% (158/258) of which were from non-relapsed patients. Most of those mutations (85.8%, 442/515) were at very low VAF (<1%) and 73.6% (379/515) of mutations were related to clonal hematopoiesis^{10, 11, 12}, including *DNMT3A*, *JAK2*, *TET2*, *ASXL1*, and *CHEK2*. These findings suggest that although a small amount of plasma samples may contain cancer-related somatic mutations which were not detected in tumor due to tumor heterogeneity, most of these mutations may come from clonal hematopoiesis.

Changes in the revised version: Lines 281-291

- And most importantly, would it be possible to use the strategy as a tumor agnostic approach so no tumor NGS data is necessary?

Reply:

We appreciate the comments from the Reviewer. As mentioned, a tumor agnostic approach has several potential advantages, such as reducing the cost of sequencing, better monitoring the tumor heterogeneity and the acquired mutations, and broadening the utility of liquid biopsy. For example, although all of patients in our study had tumor samples available, 12 of them had no detectable mutation in their tumors, thus excluding from further analysis. Therefore, using the tumor agnostic approach can extent the usage of ctDNA monitoring when tumor specimens are not available or tumor mutations are undetectable.

On the other hand, we think the main limitation to track ctDNA without tumor information is clonal hematopoiesis. In our study, we used a pool of samples from healthy donors to reduce sequencing errors and white blood cells to filter germline mutations and mutations from clonal hematopoiesis. A tumor agnostic approach should consider parallel deep sequencing of cfDNA and white blood cells to exclude clonal hematopoiesis mutations. Several approaches, such as cfDNA fragment size analysis, may also be used to filter out non-tumor-derived mutations, although further investigations are need to figure out the technical details. We added all of the discussions in the revised manuscript.

Changes in the revised version: Lines 281-291

Minor:

- What was the volume of plasma used for analysis? What was the median extracted amount of cfDNA? Did you assess purification efficiency in any way?

Reply:

The median volume of plasma we used for analysis was 4 mL (range: 1.8-6 mL) and the median extracted amount of cfDNA was 22.3 ng/mL (range: 6.8-112.7 ng/mL). We used a commercial kit (QIAmp Circulating Nucleic Acid Kit) to extract cfDNA, which offers a better cfDNA extraction efficiency compared to other commercially available extraction kits. Based on other studies¹³, the purification efficiency of this kit is over 50%.

- Please provide arrows and/or explanations for the CT images.

Reply:

We thank the reviewer for the suggestion. We added arrows to indicate the recurrence sites in the revised Fig. 3c-d and provided necessary explanations in both the figure legend and the main text.

- Please elaborate on Fig S5

Reply:

We thank the reviewer for the comment. We revised the Figure S5 (newly named Supplementary Fig. 6) with plot titles to provide a clear information of the Figure.

Reference

1. Rizopoulos D. Dynamic predictions and prospective accuracy in joint models for longitudinal and time-to-event data. *Biometrics* **67**, 819-829 (2011).
2. Huang Y, Li W, Macheret F, Gabriel RA, Ohno-Machado L. A tutorial on calibration measurements and calibration models for clinical prediction models. *Journal of the American Medical Informatics Association* **27**, 621-633 (2020).
3. Chao M, Gibbs P. Caution Is Required Before Recommending Routine Carcinoembryonic Antigen and Imaging Follow-Up for Patients With Early-Stage Colon Cancer. *Journal of Clinical Oncology* **27**, e279-e280 (2009).
4. Huang K, *et al.* Radiographic changes after lung stereotactic ablative radiotherapy (SABR)—can we distinguish recurrence from fibrosis? A systematic review of the literature. *Radiotherapy and Oncology* **102**, 335-342 (2012).
5. Ibrahim JG, Chen M-H, Sinha D. Bayesian methods for joint modeling of longitudinal and survival data with applications to cancer vaccine trials. *Statistica Sinica*, 863-883 (2004).

6. Li K, Furr-Stimming E, Paulsen JS, Luo S, Group P-HlotHS. Dynamic Prediction of Motor Diagnosis in Huntington's Disease Using a Joint Modeling Approach. *J Huntingtons Dis* **6**, 127-137 (2017).
7. Chen K, *et al.* Perioperative Dynamic Changes in Circulating Tumor DNA in Patients with Lung Cancer (DYNAMIC). *Clinical Cancer Research* **25**, 7058-7067 (2019).
8. Abbosh C, *et al.* Abstract CT023: Phylogenetic tracking and minimal residual disease detection using ctDNA in early-stage NSCLC: A lung TRACERx study.). AACR (2020).
9. Jennings LJ, *et al.* Guidelines for Validation of Next-Generation Sequencing–Based Oncology Panels: A Joint Consensus Recommendation of the Association for Molecular Pathology and College of American Pathologists. *The Journal of Molecular Diagnostics* **19**, 341-365 (2017).
10. Xie M, *et al.* Age-related mutations associated with clonal hematopoietic expansion and malignancies. *Nat Med* **20**, 1472-1478 (2014).
11. McKerrell T, *et al.* Leukemia-associated somatic mutations drive distinct patterns of age-related clonal hemopoiesis. *Cell Rep* **10**, 1239-1245 (2015).
12. Genovese G, *et al.* Clonal hematopoiesis and blood-cancer risk inferred from blood DNA sequence. *N Engl J Med* **371**, 2477-2487 (2014).
13. Diefenbach RJ, Lee JH, Kefford RF, Rizos H. Evaluation of commercial kits for purification of circulating free DNA. *Cancer genetics* **228**, 21-27 (2018).

Reviewers' Comments:

Reviewer #1:

Remarks to the Author:

I appreciate the extent of work the authors have done in this revision, the manuscript is much improved. These are my remaining concerns and remarks:

"In the future, we will recruit an independent patient cohort to validate the joint model."

The authors should note this limitation in the discussion of the manuscript.

Suppl. Fig 10b

Why does the postsurgical cox model only have 3 data points?

"The splits for training/test sets as well as reference scripts and data for model construction and evaluation were uploaded ..."

Its great that the authors have made this part of the code and data available. However, the code should reproduce figure 4a and 4b to be complete. Please add code to reproduce these figures.

"cannot make the raw targeted sequencing data publicly available due to the regulation of the Human Genetic Resources Administration of China (HGRAC). Request for accessing the raw sequencing data can be sent directly to the corresponding author if the manuscript is accepted for publication with proper HGRAC approval."

There must be a data repository within China similar to EGA or dbGap? The data should be uploaded there with appropriate terms obviously.

"we have added the statement that 'white blood cells from the buffy coat "

At which sequencing depth? For every plasma sample, or just once for each patient? These details must be included in the methods section.

Reviewer #2:

Remarks to the Author:

The revised manuscript has been greatly improved in a number of ways. The presentation of the joint model has particularly improved and it is now much more clear and promising. Some of my original concerns are, however still not thoroughly addressed. These and some additional concerns are detailed below:

- It is not sufficiently transparent how ctDNA status and disease recurrence relates. This has been nicely addressed for ctDNA positive patients with no recurrence, but needs to be similarly addressed for ctDNA negative patients with recurrence instead of simply the brief description at line 275-277.

- Furthermore, was there anything different for the tumor specific mutations investigated for patients where ctDNA is expected but missed compared to tumor specific mutations for patients where it is expected but also detected?

- Is the lead-time in recurrence detection in line with other studies in NSCLC?

- The information about collapsed sequencing depth needs to go into the main manuscript. This sequencing depth is much more important than the raw approx. 30,000X sequencing depth.

o In line with this, did the authors utilize both single and double strand consensus reads to determine mutation calls? Or were only double strand consensus reads utilized?

- Was the LOD test similarly sequenced to approx. 30,000x?

- Authors should provide more details about the ctDNA mutation calling, e.g. the collapsed read depth per sample. It would also improve the reading experience if detailed data about the individual ctDNA mutation calls was available.

I have some additional minor comments:

- Fig 2c is too difficult to interpret. Split into two figures or color code more systematically.
- Line 147. Please rephrase.
- Line 151. Rephrase, it is actually a strong prognostic biomarker.

Reviewer #1 (Remarks to the Author):

I appreciate the extent of work the authors have done in this revision, the manuscript is much improved. These are my remaining concerns and remarks:

“In the future, we will recruit an independent patient cohort to validate the joint model.”

The authors should note this limitation in the discussion of the manuscript.

Reply:

We thank the Reviewer for the suggestion. We added ‘as only internal validation was used to validate the performance of the joint model, an independent patient cohort is warranted to further validate the joint model in the future’ in the discussion section.

Changes in the revised version (clean version): Lines 350-351

Suppl. Fig 10b

Why does the postsurgical cox model only have 3 data points?

Reply:

We deeply thank the Reviewer for pointing out this error in our previous submission, and we apologize for the incorrect data of postsurgical Cox model training datasets. We re-examined the code and found that the ‘aucJM’ function (from the ‘jmbayes’ package) that we used in the postsurgical Cox model analysis could not properly handle ‘NA’ values. Five patients in our cohort had no postsurgical blood samples, resulting in ‘NA’ data in the postsurgical status. These ‘NA’ values messed up the aucJM function and produced incorrect results in previous supplementary Fig 10. As a result, only three training datasets without ‘NA’ data correctly generated results during the cross-validation stratification (as shown in the previous supplementary Fig 10b). Therefore, we modified the code to exclude patients without postsurgical blood samples when analyzing the postsurgical Cox model and revised the related figures and tables as shown below (Fig 4b, Supplementary Fig 10, Supplementary Fig. 12, and Supplementary Table 6). All the renewed codes were also available at https://github.com/cancer-oncogenomics/ctDNA-dynamic-prediction-lung-cancer/blob/main/function/cross_validation.R (line #115-#117). Also, this correction of codes and data did not influence our conclusion, that is, the joint model has a superior ability to predict recurrence status at 12 and 15 months post-surgery, comparing with the static Cox model using only postsurgical ctDNA status or the landmarking ctDNA status Cox model .

Of note, we also carefully checked all of the other codes that were used in the manuscript, and we confirmed that the results of other models, which are, the joint model and the landmark Cox model, were correct and were not affected by the ‘NA’ data issues as data of at least one or more other time points were available in the other two model.

Figure 4b The comparison of model performance of the testing dataset between the joint model and Cox models.

Supplementary Fig. 10 The comparison of model performance of training datasets between the joint model and Cox models.

Supplementary Fig. 12 The comparison of model performance between the joint model and Cox models using only AD patients.

“The splits for training/test sets as well as reference scripts and data for model construction and evaluation were uploaded ...”

Its great that the authors have made this part of the code and data available. However, the code should reproduce figure 4a and 4b to be complete. Please add code to reproduce these figures.

Reply:

We thank the Reviewer for the valuable suggestions and apologize for the unclear annotations of the code. Figure 4a is a schematic diagram and the line plots in the figure were drawn using Adobe Illustration, so there is no code available for Figure 4a. The code that reproduces Figure 4b was already included in the ‘run_joint_model_analysis.R’ file (available at https://github.com/cancer-oncogenomics/ctDNA-dynamic-prediction-lung-cancer/blob/main/run_joint_model_analysis.R) of the previous submission. The details of the code were also shown as follows:

```
## plot the CV results
```

```
# CV_all_patients/JMvsCox_testing.pdf ~ Fig 4b; CV_all_patients/JMvsCox_training.pdf ~ Supplementary Fig 10;
```

```
# CV_all_patients/betweenJMs.pdf ~ Supplementary Fig 15
```

```
plot_evaluation('results/CV_all_patients', out_all)
```

This section of codes (line #62) produces several graph files including Figure 4b, Supplementary Fig 10, and Supplementary Fig 15. We added further annotations in the code to specify the codes that reproduce the figures.

```
## plot the personalized prediction ~ Fig 4c,d; Supplementary Fig 13
```

```
...
```

This section of codes (line #67 - line #105) produces Fig 4c and 4d.

“cannot make the raw targeted sequencing data publicly available due to the regulation of the Human Genetic Resources Administration of China (HGRAC). Request for accessing the raw sequencing data can be sent directly to the corresponding author if the manuscript is accepted for publication with proper HGRAC approval.”

There must be a data repository within China similar to EGA or dbGap? The data should be uploaded there with appropriate terms obviously.

Reply:

We appreciate the comments from the Reviewer and understand his/her concerns. Uploading the sequencing data to a data repository within China still needs approval from the Human Genetics Resources Administration of China (HGRAC). We have submitted the data uploading request to HGRAC in early April this year, and our application is currently under review. As soon as we receive the approval from HGRAC, which is likely to be this late June, we will upload all of our raw sequencing data to a public data repository and notify your office of the access number when the uploading process is completed.

“we have added the statement that ‘white blood cells from the buffy coat “

At which sequencing depth? For every plasma sample, or just once for each patient? These details must be included in the methods section.

Reply:

We thank the Reviewer for his/her valuable suggestions and apologize for the missing information. Only one white blood cells (WBC) sample from the buffy coat after plasma preparation at baseline for each patient was analyzed as germline control. The mean sequencing coverage depth of the WBC sample was ~300×, and germline mutations from all the consecutive plasma samples of the same patient were filtered out using the WBC sequencing data of this patient at baseline. We added ‘white blood cells from the buffy coat after plasma preparation were also collected from the same patient at baseline’ and ‘the mean sequencing coverage depth of the white blood cells was ~300×’ in the revised manuscript.

Changes in the revised version (clean version): Lines 371-372

Reviewer #2 (Remarks to the Author):

The revised manuscript has been greatly improved in a number of ways. The presentation of the joint model has particularly improved and it is now much more clear and promising. Some of my original concerns are, however still not thoroughly addressed. These and some additional concerns are detailed below:

- It is not sufficiently transparent how ctDNA status and disease recurrence relates. This has been nicely addressed for ctDNA positive patients with no recurrence, but needs to be similarly addressed for ctDNA negative patients with recurrence instead of simply the brief description at line 275-277.

Reply:

We appreciate the comments from the Reviewer and understand his/her concerns. There were seven relapsed patients who were negative for plasma ctDNA before recurrence. One patient (P038) only had brain metastasis upon recurrence, considering that the blood-brain barrier may potentially prevent the release of its ctDNA into the blood circulation¹, it may cause negative ctDNA detection from peripheral blood plasma. Patients P024 and P079 had no detectable ctDNA even at baseline, and the VAF of pretreatment ctDNA in patient P037 was as low as 0.04%, suggesting that the tumors in these patients may not actively release ctDNA. Other reasons for a negative ctDNA detection could be due to the heterogeneity of the primary tumor, that is, the seeding clones of the relapsed tumor might not be included in the sequenced region of the tumor biopsy and the mutated genes of the relapsed tumor were not covered by our testing panel. We have added the above discussions into the revised manuscript.

Changes in the revised version (clean version): Lines 276-285

- Furthermore, was there anything different for the tumor specific mutations investigated for patients where ctDNA is expected but missed compared to tumor specific mutations for patients where it is expected but also detected?

Reply:

We appreciate the comments from the Reviewer. There was no significant difference between tumor-specific mutations in patients where ctDNA is expected but missed and patients where it is expected and detected. *TP53* were the most frequent mutated gene in patients where ctDNA is expected but missed (6/7). However, there was no significant difference in *TP53* mutation frequency between patients whose ctDNA is expected but missed and patients whose ctDNA is expected and detected (6/7 vs. 4/27; p=1; Fisher's exact). Similarly, for all other mutated genes that occurred in patients where ctDNA is expected but missed (n=7) or patients where ctDNA is expected but detected (n=27), no difference in mutation frequency between the two groups was detected, such as *EGFR* (2/7 vs. 8/27; p=1; Fisher's exact), *KRAS* (2/7 vs. 2/27; p=0.18; Fisher's exact) *KEAPI* (2/7 vs 4/27; p=0.58; Fisher's exact) and *LRP1B* (2/7 vs 4/27; p=1; Fisher's exact). Furthermore, the coverage depth of those 'expected but missed' mutations was not lower than that of the 'expected and detected' mutations (p=1; Wilcoxon signed rank test). Overall, we think the detection power of mutations at different genomic positions was similar and different tumor-specific mutations may not affect the ctDNA detection. We think the activity of ctDNA release and clearance, the site of metastasis, and tumor heterogeneity were likely to be the major factors that affect the posttreatment ctDNA detection.

Changes in the revised version (clean version): Lines 285-287

- Is the lead-time in recurrence detection in line with other studies in NSCLC?

Reply:

We appreciate the comments from the Reviewer. Although most of the previous NSCLC studies demonstrated that ctDNA detection preceded radiological recurrence, the lead times were not consistent among these studies. The specific clinical characteristics in each study cohort (e.g., pathological stage and treatment strategy) may influence the

power of ctDNA preceding radiological recurrence. Abbosh *et al.* reported a median lead time of 70 days of ctDNA in resected NSCLC² with a three-month repeated blood collection period, which was similar to both the blood collection period (3 month) and median lead time (88 days) in our study. In contrast, Chen *et al.* conducted another resected NSCLC study and they reported a longer median lead time of 165 days³. However, in Chen *et al.*'s study, 65.4% of patients were stage III or above, compared with that of 49% in our cohort, and their cohort size was relatively small (n=26). Chaudhuri *et al.*⁴ reported a median lead time of 5.2 months with a three to six-month repeated blood collection period in localized lung cancer where most of the patients (88%, 35/40) were treated with radiotherapy, as compared with the surgical resection in our study. Similarly, Moding *et al.*⁵ reported a lead time of 4.1 months in locally advanced NSCLC where patients were treated with consolidation immune checkpoint inhibition. Unfortunately, our study was also partially affected by the COVID-19 pandemic. A total of 11 patients in our cohort were delayed on their arranged blood drawn time point, and therefore had their blood samples collected right before or at the time of recurrence. As a result, the lead times in these patients may be largely underestimated. We have added the above discussions in the revised manuscript.

Changes in the revised version (clean version): Lines 290-302

- The information about collapsed sequencing depth needs to go into the main manuscript. This sequencing depth is much more important than the raw approx. 30,000X sequencing depth.

o In line with this, did the authors utilize both single and double strand consensus reads to determine mutation calls? Or were only double strand consensus reads utilized?

Reply:

We thank the Reviewer for the valuable suggestions. We added the information about the collapsed consensus sequencing depth in the revised manuscript. As for mutation calling, we utilized both single- and double-stranded consensus reads in this tumor-guided study. As shown by previous researches, one of the limitations of UMI method is the low recovery rate of duplex sequences⁶, so some tumor-informed mutations were only identified in single-stranded consensus reads. Therefore, in order to improve the sequencing coverage, we used both single- and double-stranded consensus reads to call mutations.

Changes in the revised version (clean version): Lines 460-461

- Was the LOD test similarly sequenced to approx. 30,000x?

Reply:

The LOD test used the same sequencing procedure to a sequencing depth of ~30,000× as the clinical samples. In the revised manuscript, we have modified this part of the description as follows: 'as tested with ~30,000× deep sequencing of DNA mixtures of two reference human DNA samples'.

Changes in the revised version (clean version): Lines 407-408

- Authors should provide more details about the ctDNA mutation calling, e.g. the collapsed read depth per sample. It would also improve the reading experience if detailed data about the individual ctDNA mutation calls was available.

Reply:

As suggested, we have added more details about ctDNA mutation calling in the revised Method section. Furthermore, in the newly added Supplementary Table 8, we provided detailed information about DNA mutation calling of each plasma sample, including the mean collapsed consensus read depth and plasma DNA amount. Also, we added the

sequencing depth at the specified genomic position in plasma DNA in the Supplementary Table 2 for better reading experience.

Changes in the revised version (clean version): Lines 455-460

I have some additional minor comments:

- Fig 2c is too difficult to interpret. Split into two figures or color code more systematically.

Reply:

We thank the Reviewer for his/her valuable suggestions. We have revised the color code of Fig 2c and added additional descriptions at the end of each line as shown below to help better interpret the figure.

Fig. 2c Kaplan-Meier curve of RFS in stage II-III patients stratified by both ACT treatment and postsurgical ctDNA status

- Line 147. Please rephrase.

Reply:

In the revised manuscript, we rephrased the sentence as follows: ‘... , suggesting that ctDNA status may have potential clinical values in evaluating the effectiveness of ACT.’.

Changes in the revised version (clean version): Lines 145-146

- Line 151. Rephrase, it is actually a strong prognostic biomarker.

Reply:

We thank the Reviewer for the suggestion. We revised it to ‘Taken together, these results demonstrate that MRD detected by plasma ctDNA after definitive therapy is a promising prognostic biomarker for resectable NSCLC patients.’

Changes in the revised version (clean version): Lines 150

1. Bettegowda C, *et al.* Detection of circulating tumor DNA in early- and late-stage human malignancies. *Sci Transl Med* **6**, 224ra224-224ra224 (2014).
2. Abbosh C, *et al.* Phylogenetic ctDNA analysis depicts early-stage lung cancer evolution. *Nature* **545**, 446-451 (2017).
3. Chen K, *et al.* Perioperative Dynamic Changes in Circulating Tumor DNA in Patients with Lung Cancer (DYNAMIC). *Clinical Cancer Research* **25**, 7058-7067 (2019).
4. Chaudhuri AA, *et al.* Early Detection of Molecular Residual Disease in Localized Lung Cancer by Circulating Tumor DNA Profiling. *Cancer Discov* **7**, 1394-1403 (2017).
5. Moding EJ, *et al.* Circulating tumor DNA dynamics predict benefit from consolidation immunotherapy in locally advanced non-small-cell lung cancer. *Nature Cancer* **1**, 176-183 (2020).
6. Wang TT, *et al.* High efficiency error suppression for accurate detection of low-frequency variants. *Nucleic Acids Research* **47**, e87-e87 (2019).

Reviewers' Comments:

Reviewer #1:

Remarks to the Author:

The authors have addressed all my remaining concerns.

It is imperative that the authors work towards sharing their data with approval from HGRAC, to ensure their work is fully transparent and reproducible.

There must be a data repository within China similar to EGA or dbGap? The data should be uploaded there with appropriate terms obviously.

Reply:

We appreciate the comments from the Reviewer and understand his/her concerns. Uploading the sequencing data to a data repository within China still needs approval from the Human Genetics Resources Administration of China (HGRAC). We have submitted the data uploading request to HGRAC in early April this year, and our application is currently under review. As soon as we receive the approval from HGRAC, which is likely to be this late June, we will upload all of our raw sequencing data to a public data repository and notify your office of the access number when the uploading process is completed.

Reviewer #2:

Remarks to the Author:

The authors have addressed all outstanding concerns. The increase in mutation calling details have done well for the manuscript in general.

Reviewer #3:

Remarks to the Author:

This manuscript compares the detection of cell free tumor (ct) DNA in blood samples of lung cancer patients prior to and at different times subsequent to surgery, with or without adjuvant chemotherapy (ACT). The conclusions are that in stage II-III lung cancer patients, the detection of post-surgical or post ACT ctDNA was associated with worse relapse free survival. In addition, the application of a biostatistical algorithm developed by the authors made it possible to detect recurrence and treatment failure somewhat sooner. How clinically useful this may be, however, is not adequately addressed. A more useful approach might be in the neoadjuvant setting, which has become more common clinically, where a major reduction in tumor ctDNA at the time of surgery might be tested as a predictor of adjuvant therapy efficacy. Other concerns are as follows:

1. Therapy for lung tumors is rapidly evolving. For example, the clinical benefit of neoadjuvant immuno-, chemo- or biologically targeted therapy for patients prior to surgery is becoming standard and/or being systematically evaluated as are adjuvant chemo-vs immunotherapy with immunotherapy already approved as adjuvant therapy for certain forms of lung cancer. Thus, results in this manuscript may be somewhat dated, and the authors' findings might be much more relevant to present day therapies some of these variables were analyzed. These issues detract from the usefulness of any clinical conclusions that can drawn.

2. There are also several published studies testing ctDNA methodology in lung cancer as well as other tumor types. The authors analyze relapse free survival in lung cancers and break out data for lung AD, but results with squamous ca would be of interest for comparison.

3. While Nat. Comm. has a broad scientific audience, this manuscript is primarily a clinical study, which utilizes what are now relatively standard approaches for ctDNA detection. Thus, it seems that this manuscript would find its most appropriate audience in a specialized clinical oncology journal. For example, most of the results involve comparisons of different biostatistical methods. and meaningful evaluation of the results may require a level of biostatistical expertise not available or not of interest to the general readership of Nat. Comm.

Other points:

1. The authors report 49% ctDNA positivity of untreated adenocarcinoma (AD) compared to 100% for other lung cancer subtypes. Yet, they indicate that a similar fraction (20%) of both AD and squamous ca patients remained positive after surgery. The authors need to clarify whether these positive are subsets of the entire patient group or reflect only the subset in the case of AD of those positive at the time of surgery.

Reviewer #4:

Remarks to the Author:

Original paper describing a significant correlation between MRD+ status and recurrence risk in surgically resected lung cancer patients. The clinical validation of ctDNA as adjuvant biomarker is an hot topic for clinical lung cancer research, aiming to stratify patients' prognosis and select best candidates to adjuvant therapies. Although limited by small sample size and patients' heterogeneity the findings of this study provide an interesting contribution to the current scientific debate, supporting the clinical utility of MRD assessment which need to be validated in the context of prospective clinical trials.

Few additional suggestion to improve the quality of the paper:

Please detail a bit more the clinical background of this research, explaining current clinical needs in the adjuvant setting, including heterogeneous survival rates associated to the different TNM staging, limited benefit associated to chemo, high overtreatment rate with related futile toxicities, absence of reliable prognostic/predictive biomarkers for clinical use, etc...

Consider to report and discuss also TRACERx data recently presented by Abbosh at AACR 2020 meeting (CT023 - Phylogenetic tracking and minimal residual disease detection using ctDNA in early-stage NSCLC: A lung TRACERx study)

"For patients clinically defined as low-risk populations, if they show a strong ctDNA 157 positive status, they may benefit from additional ACT treatment". This is supposed to be the research hypothesis. Please define the "low-risk population" and edit the following sentences detailing a bit more the aims of the analysis. This analysis has several limitations related to the very low number of patients who do not received ACT. Please discuss potential implications on the results.

"Our results demonstrated that detection of tumor-specific mutations in ctDNA samples 184 at any longitudinal time points was more significantly associated with worse RFS (HR, 8.5; 95% 185 CI, 3.7-20; $p < 0.001$; Fig. 2d) compared to only using the postsurgical ctDNA status (Fig. 2a)". It's not clear how did you compare longitudinal versus post-surgical DNA ? what do you mean as "more significantly associated" ? This looks just as a simple comparison between positive versus negative ctDNA longitudinal analysis. What do you mean as "any longitudinal timepoints" ? please detail in order to support the clinical utility of this data.

In light of the small sample size and patients' heterogeneity, the results of this work need to be considered preliminary and require to be confirmed in the context of prospective independent, external, and larger cohorts of patients. This concept should be clearly stated in the text as well as in the abstract conclusion.

Some typos in text to be edited.

Reviewer #1 (Remarks to the Author):

The authors have addressed all my remaining concerns.

It is imperative that the authors work towards sharing their data with approval from HGRAC, to ensure their work is fully transparent and reproducible.

There must be a data repository within China similar to EGA or dbGap? The data should be uploaded there with appropriate terms obviously.

Reply:

We appreciate the comments from the Reviewer and understand his/her concerns. Uploading the sequencing data to a data repository within China still needs approval from the Human Genetics Resources Administration of China (HGRAC). We have submitted the data uploading request to HGRAC in early April this year, and our application is still under review. Based on the latest updates from HGRAC, the final decision from HGRAC might be delayed to September this year due to their excessive workload. As soon as we receive the approval from HGRAC, we will upload all of our raw sequencing data to a public data repository and notify the journal of the access number when the uploading process is completed.

Reviewer #2 (Remarks to the Author):

The authors have addressed all outstanding concerns. The increase in mutation calling details have done well for the manuscript in general.

Reviewer #3 (Remarks to the Author):

This manuscript compares the detection of cell free tumor (ct) DNA in blood samples of lung cancer patients prior to and at different times subsequent to surgery, with or without adjuvant chemotherapy (ACT). The conclusions are that in stage II-III lung cancer patients, the detection of post-surgical or post ACT ctDNA was associated with worse relapse free survival. In addition, the application of a biostatistical algorithm developed by the authors made it possible to detect recurrence and treatment failure somewhat sooner. How clinically useful this may be, however, is not adequately addressed. A more useful approach might be in the neoadjuvant setting, which has become more common clinically, where a major reduction in tumor ctDNA at the time of surgery might be tested as a predictor of adjuvant therapy efficacy. Other concerns are as follows:

1. Therapy for lung tumors is rapidly evolving. For example, the clinical benefit of neoadjuvant immuno-, chemo- or biologically targeted therapy for patients prior to surgery is becoming standard and/or being systematically evaluated as are adjuvant chemo-vs immunotherapy with immunotherapy already approved as adjuvant therapy for certain forms of lung cancer. Thus, results in this manuscript may be somewhat dated, and the authors' findings might be much more relevant to present day therapies some of these variables were analyzed. These issues detract from the usefulness of any clinical conclusions that can draw.

Reply:

We appreciate the comments from the Reviewer and understand his/her concerns. We agree that the newly developing therapeutic strategies for lung cancer, such as neoadjuvant therapy or adjuvant immunotherapy, have shown their potential to improve the survival of patients and have prompted substantial interest. In the most recent version of NCCN guidelines, adjuvant chemotherapy was the major treatment option after direct surgery. We think adjuvant

chemotherapy would still be one of the major options for operable NSCLC patients for the foreseeable future, owing to its lower cost compared to adjuvant immunotherapy. Moreover, this study was a clinical trial that was first initiated in 2018. It was designed to investigate the utility of ctDNA in post-surgery/ACT disease monitor and the potential of postsurgical ctDNA to guide adjuvant chemotherapy. Our findings can provide insights for a more effective adjuvant chemotherapy strategy and post-operation monitoring. Furthermore, the methodologies used in this study, such as deep next-generation sequencing and joint modeling, are not limited to adjuvant chemotherapy, and they can be easily expanded to neoadjuvant settings in future studies. Further studies of our methodologies in neoadjuvant chemotherapy are under consideration. We added these in the discussion section of the manuscript.

Changes in the revised version (clean version): Lines 421-425

2. There are also several published studies testing ctDNA methodology in lung cancer as well as other tumor types. The authors analyze relapse free survival in lung cancers and break out data for lung AD, but results with squamous ca would be of interest for comparison.

Reply:

We agree with the Reviewer that it is worth investigating both AD and squamous cell carcinoma (SqCC), as 37% of the patients in our study were diagnosed with SqCC. Therefore, we re-analyzed the results and revised the manuscript as follows: Firstly, as described in the manuscript, we compared the detection rate of pretreatment ctDNA and found that AD patients had a much lower detection rate than SqCC patients. Although different histology (SqCC vs AD) showed no difference in relapse-free survival ($p=0.52$), SqCC patients showed significantly better RFS than AD patients when only considering pretreatment ctDNA positive patients (HR: 0.45; 95% CI: 0.2-0.99; $p<0.05$). We added this result in the revised manuscript. Secondly, we analyzed the association between postsurgical/post-ACT/longitudinal ctDNA status and relapse-free survival and demonstrated that the prognostic value of ctDNA status. These analyses were also performed on both AD and SqCC sub-cohorts and the results were shown in Supplementary Fig. 5. The postsurgical/longitudinal ctDNA status was associated with worse recurrence-free survival in both AD and SqCC sub-cohorts with similar hazard ratios (postsurgical ctDNA: 4.19 vs 3.89; longitudinal ctDNA: 8.33 vs 8.56). Furthermore, we compared the ctDNA-positive detection rate of relapsed patients as well as ctDNA lead time in AD and SqCC patients, and no statistical difference was found between the two sub-cohorts ($p=1$ and $p=0.92$, respectively).

Furthermore, as mentioned by the Reviewer, there were several published studies testing ctDNA methodology in lung cancer as well as other cancers, including a recent study investigating the prognostic value of ctDNA and its association with adjuvant treatment in stage III colon cancer; however, no studies focused on ctDNA status guiding ACT in lung cancer. In this study, we showed pioneering work of using postsurgical ctDNA status to guide ACT and applying joint modeling to dynamically predict personalized recurrence risk.

Changes in the revised version (clean version): Lines 137-140, 157, 232

3. While Nat. Comm. has a broad scientific audience, this manuscript is primarily a clinical study, which utilizes what are now relatively standard approaches for ctDNA detection. Thus, it seems that this manuscript would find its most appropriate audience in a specialized clinical oncology journal. For example, most of the results involve comparisons of different biostatistical methods. and meaningful evaluation of the results may require a level of biostatistical expertise not available or not of interest to the general readership of Nat. Comm.

Reply:

We thank the Reviewer's comments. Based on the research scope published on *Nature Communications*' website, clinical researches are also one of the main interests of *Nature Communications*. Also, a significant amount of the audience of Nat. Comm are clinical researchers. Therefore, we believe our research is likely to attract audiences of *Nature Communications* who are interested in clinical researches, lung cancer treatment, risk prediction, and biomarker researches.

Other points:

1. The authors report 49% ctDNA positivity of untreated adenocarcinoma (AD) compared to 100% for other lung cancer subtypes. Yet, they indicate that a similar fraction (20%) of both AD and squamous ca patients remained positive after surgery. The authors need to clarify whether these positive are subsets of the entire patient group or reflect only the subset in the case of AD of those positive at the time of surgery.

Reply:

We thank the Reviewer for the comments and we apologize for the confusion. Of the 85 patients who had available postsurgical plasma samples, 51 were AD and 31 were squamous cell carcinoma. 19.6% (10/51) of AD patients and 19.4% (6/31) of SqCC patients were postsurgical ctDNA positive, which both took the entire patient number as the base. We revised this part as ‘Although different histology subtypes have distinct ctDNA detection rates in presurgical plasma samples (Supplementary Fig. 3b), there was no difference ($p=1.0$) in the postsurgical ctDNA positive rate between AD (19.2%; 10 out of all the 52 AD patients) and SqCC (19.4%, 6 out of all the 31 SqCC patients).’ in the manuscript to avoid misunderstandings.

Changes in the revised version (clean version): Lines 149-150

Reviewer #4 (Remarks to the Author):

Original paper describing a significant correlation between MRD+ status and recurrence risk in surgically resected lung cancer patients. The clinical validation of ctDNA as adjuvant biomarker is an hot topic for clinical lung cancer research, aiming to stratify patients’ prognosis and select best candidates to adjuvant therapies. Although limited by small sample size and patients’ heterogeneity the findings of this study provide an interesting contribution to the current scientific debate, supporting the clinical utility of MRD assessment which need to be validated in the context of prospective clinical trials.

Few additional suggestion to improve the quality of the paper:

Please detail a bit more the clinical background of this research, explaining current clinical needs in the adjuvant setting, including heterogeneous survival rates associated to the different TNM staging, limited benefit associated to chemo, high overtreatment rate with related futile toxicities, absence of reliable prognostic/predictive biomarkers for clinical use, etg...

Reply:

We thank the Reviewer for the constructive suggestion. We revised the manuscript and added more clinical background of adjuvant chemotherapy to emphasize the clinical needs of the development of biomarkers for adjuvant chemotherapy.

Changes in the revised version (clean version): Lines 52-59

Consider to report and discuss also TRACERx data recently presented by Abbosh at AACR 2020 meeting (CT023 - Phylogenetic tracking and minimal residual disease detection using ctDNA in early-stage NSCLC: A lung TRACERx study)

Reply:

We thank the Reviewer for the valuable suggestion. Abbosh *et al.*'s 2020 study was mentioned in the discussion part of the manuscript. Furthermore, we added more discussions about their methodology and results in the revised manuscript.

Changes in the revised version (clean version): Lines 383-385, 410-415

“For patients clinically defined as low-risk populations, if they show a strong ctDNA 157 positive status, they may benefit from additional ACT treatment”. This is supposed to be the research hypothesis. Please define the “low-risk population” and edit the following sentences detailing a bit more the aims of the analysis. This analysis has several limitations related to the very low number of patients who do not received ACT. Please discuss potential implications on the results.

Reply:

We thank the Reviewer for pointing out the inaccurate description of the research hypothesis in our initial submission. All of the patients involved in the analysis had stage II or III disease, so they should be considered as ‘high-risk’. Our research aim is to identify patients who are less likely to benefit from ACT within these ‘high-risk’ patients, thus minimizing futile toxicities and overtreatments. The definition of the ‘high-risk’ population was added in the revised manuscript. We also rephrased our research hypothesis as follows: ‘We hypothesized that those patients who were clinically defined as high-risk populations but had no detectable postsurgical ctDNA may not benefit from additional ACT treatment’. We agreed that our result was preliminary due to the low number of patients. Moreover, this result used only stage II-III patients who were considered as high-risk. The application of ctDNA to guide ACT on ‘low-risk’ patients (patients were not recommended for ACT in current clinical settings) requires further studies. We added this discussion in the revised manuscript.

Changes in the revised version (clean version): Lines 186-192, 443-445,465-468

“Our results demonstrated that detection of tumor-specific mutations in ctDNA samples 184 at any longitudinal time points was more significantly associated with worse RFS (HR, 8.5; 95% CI, 3.7-20; $p < 0.001$; Fig. 2d) compared to only using the postsurgical ctDNA status (Fig. 2a)”. It’s not clear how did you compare longitudinal versus post-surgical DNA ? what do you mean as “more significantly associated” ? This looks just as a simple comparison between positive versus negative ctDNA longitudinal analysis. What do you mean as “any longitudinal timepoints” ? please detail in order to support the clinical utility of this data.

Reply:

We appreciate the comments from the Reviewer and we apologize for the confusion in our previous submission. For ‘any longitudinal timepoints’, we meant any time point(s) during posttreatment surveillance. In Fig 2d, we compared RFS between longitudinal positive patients (patients with at least one positive ctDNA during posttreatment surveillance) and longitudinal negative patients (patients whose ctDNA were always negative during posttreatment surveillance). We revised the sentence as “Our results showed that patients with detectable ctDNA at any time point(s) during posttreatment surveillance had significantly lower DFS than those who always had negative ctDNA detection after surgery (HR, 8.5; 95% CI, 3.7-20; $p < 0.001$; Fig. 2d). In contrast, postsurgical ctDNA positivity was associated with worse RFS than negativity with a lower hazard ratio (HR: 4.0; 95% CI: 2.0-8.0; $p < 0.001$; Fig. 2a).”.

To further emphasize the advantage of using longitudinal plasma monitoring over only using postsurgical ctDNA monitoring in disease monitor, we added ‘79% (27/34) of the relapsed patients detected at least one positive ctDNA during disease surveillance, comparing with only 41% (14/34) of recurrence in patients with positive postsurgical ctDNA. This indicated that longitudinal ctDNA monitoring had additional clinical values of identifying more relapsed patients than postsurgical ctDNA monitoring.’ in the revised manuscript.

Changes in the revised version (clean version): Lines 224-231

In light of the small sample size and patients' heterogeneity, the results of this work need to be considered preliminary and require to be confirmed in the context of prospective independent, external, and larger cohorts of patients. This concept should be clearly stated in the text as well as in the abstract conclusion.

Reply:

We thank the Reviewer for the valuable suggestions, and we agreed with the Reviewer that some of our results were preliminary and additional studies were needed to further confirm our conclusions. In the revised manuscript, we have added additional discussions of the limitation of our study in both the main text and the abstract conclusion section.

Changes in the revised version (clean version): Lines 34-36, 465-468

Some typos in text to be edited.

Reply:

We thank the Reviewer for the comments. We have re-edited the manuscript to correct all the typos.

1. Choi M, Shi J, Zhu Y, Yang R, Cho K-H. Network dynamics-based cancer panel stratification for systemic prediction of anticancer drug response. *Nature Communications* **8**, 1940 (2017).

Reviewers' Comments:

Reviewer #3:

Remarks to the Author:

The authors have responded satisfactorily to the criticisms raised, and the revised manuscript is now acceptable for publication

Reviewer #4:

Remarks to the Author:

The authors have addressed the majority of the pointed out limitations.